# Narcolepsy risk loci outline role of T cell autoimmunity and infectious triggers in narcolepsy

Narcolepsy type 1 (NT1) is caused by a loss of hypocretin/orexin transmission. Risk factors include pandemic 2009 H1N1 influenza A infection and immunization with Pandemrix®. Here, we dissect disease mechanisms and interactions with environmental triggers in a multi-ethnic sample of 6,073 cases and 84,856 controls. We fine-mapped GWAS signals within HLA (DQ0602, DQB1*03:01 and DPB1*04:02) and discovered seven novel associations (CD207, NAB1, IKZF4-ERBB3, CTSC, DENND1B, SIRPG, PRF1). Significant signals at TRA and DQB1*06:02 loci were found in 245 vaccination-related cases, who also shared polygenic risk. T cell receptor associations in NT1 modulated TRAJ*24, TRAJ*28 and TRBV*4-2 chain-usage. Partitioned heritability and immune cell enrichment analyses found genetic signals to be driven by dendritic and helper T cells. Lastly comorbidity analysis using data from FinnGen, suggests shared effects between NT1 and other autoimmune diseases. NT1 genetic variants shape autoimmunity and response to environmental triggers, including influenza A infection and immunization with Pandemrix®.

The sleep disorder type 1 narcolepsy (NT1) affects ~0.03% individuals across ethnic groups and populations[1–3] and onset manifestation most commonly occurs in childhood or adolescence. As one of the symptoms, cataplexy, is almost entirely specific to NT1, diagnosis is clinical, although sleep recordings are performed for confirmation. Symptoms are caused by the autoimmune destruction of hypocretin/orexin (HCRT) neurons in the hypothalamus[4]. NT1 is 97% associated with HLA-*DQA1*01:02- DQB1*06:02*, alleles encoding the DQ heterodimer DQ0602[5,6]. Other predisposing loci include T-cell receptor (TCR) loci *TRA* and *TRB*, a type 1 interferon response receptor gene, *IFNAR1*, as well as other autoimmune-associated genes (*CTSH, P2RY11, ZNF365,* and *TNFSF4*)[7–10]. Recent studies identified HCRT, notably C-amidated fragments of secreted HCRT peptides (HCRT_{NH2}), as CD4[+] T-cell autoantigens[11–15].

Triggers of NT1 autoimmunity point to Influenza-A[9,16,17] and, secondarily, *Streptococcus pyogenes* infections[18,19]. Onset in children is abrupt and seasonal peaking between spring and summer[16], presumably following a winter infection. Further, multiple countries have reported increased incidence of NT1 4–6 months following the

2009 H1N1 (pH1N1) "swine flu" pandemic[16,17,20]. Finally, immunization with Pandemrix®, a pH1N1 vaccine created to prevent the 2009 pandemic, is an established trigger for NT1[17,20,21]. Increased incidence following Pandemrix® was first seen in Northern Europe, with incidence in children increasing from 0.79/100,000 to 6.3/100,000[21]. Specificity is striking, as increased NT1 was later detected in all European countries where Pandemrix® was used, whereas countries using other vaccine brands did not display vaccination-associated increases in incidence[17,20]. The reason for the vaccine brand specificity may involve differences in flu antigen preparations and/or timing of vaccination when infections peaked in some countries[17,20]. Frequency of other autoimmune diseases did not increase following Pandemrix® vaccination[20].

In this study, we characterize novel genetic factors for NT1 across multiple ethnic groups, performing computational and functional fine mapping. Our findings establish a compelling pathophysiological mechanism for the disease that implicate antigen presentation by DQ0602 to specific CD4[+] T cells and subsequent CD8[+] T-cell activation, with applications in the autoimmune disease and vaccination fields.

✉ e-mail: mignot@stanford.edu

## Results

### GWAS discovers novel risk loci for NT1

5,848 cases and 61,153 controls derived from ten cohorts were used as the initial discovery GWAS sample (Table S1). We found associations in HLA ($P < 10^{-216}$), confirmed previously identified loci *(TRA, TRB, CTSH, IFNAR1, ZNF365, TNFSF4)* and found 7 novel loci near *CD207, NAB1, IKZF4-ERBB3, CTSC, DENND1B, SIRPG* and *PRF1* (Figs. 1A and 2A, B, Table 1; Supplementary Figs. 1 and 2). We observed that most associations were shared across all ethnic groups. Significance between-cohort heterogeneity was observed with *TRA, SIRPG* and *DENND1B* (Table S2). Finally, as both influenza infections and, in rare cases, immunization with Pandemrix®, associates with NT1[20], 245 vaccination induced NT1 cases identified in four countries were also studied. In this sub-sample, we found GWAS significant signals with HLA-DQB1*06:02 and *TRA* rs1154155, as well as shared polygenetic risk (Table 1 and Supplementary Fig. 3). The lack of association of other loci is likely due to the small number of individuals with vaccination-related narcolepsy.

Interestingly, GWAS results are unusually rich in missense variants. In addition to HLA polymorphisms, these include a TRAJ24 (F8V) substitution, polymorphisms in langerin (CD207 N288D and K313I), as well as variants in CTSC (I453L) and PRF1 (A91V); the last two are known hypomorphs involved in autosomal recessive conditions with abnormal sensitivity to viral infections. Finally, we found that the effects of some of these variants colocalized between NT1 and type 1 diabetes (CTSH G11R and SIRPG S286L, posterior probability = 1.0). Functional effects of these missense variants are detailed in Supplementary Data 1 and Supplementary Fig. 2.

Other polymorphisms found in the GWAS and associated with other autoimmune diseases include *ZNF365* (atopic eczema, ankylosing spondylitis, Crohn's disease, psoriasis, primary sclerosing cholangitis, ulcerative colitis, posterior probability = 1.0), *TNFSF4* (eczema, asthma and allergic diseases, posterior probability = 1.0), *NAB1* (primary biliary cholangitis $r^2 = 0.48$, rheumatoid arthritis, $r^2 = 0.15$) and *IKZF4-ERBB* (vitiligo and alopecia areata $r^2 = 0.36$) (see Table Supplementary Data 1 and Supplementary Fig. 2 for functional descriptions and associations).

### NT1 shares variants with other autoimmune diseases

Heritability in NT1 is similar to other pediatric autoimmune diseases[22]; GCTA estimated observed scale heritability to be h2$_{SNP}$[ci] = 0.403 [0.015]. Using a prevalence of 0.03%[1,3], we estimate population heritability at h2$_{SNP}$[ci] = 0.231 [0.0088], consistent with twin studies[23]. We found that shared heritability was largest with sleepiness and daytime napping and with autoimmune

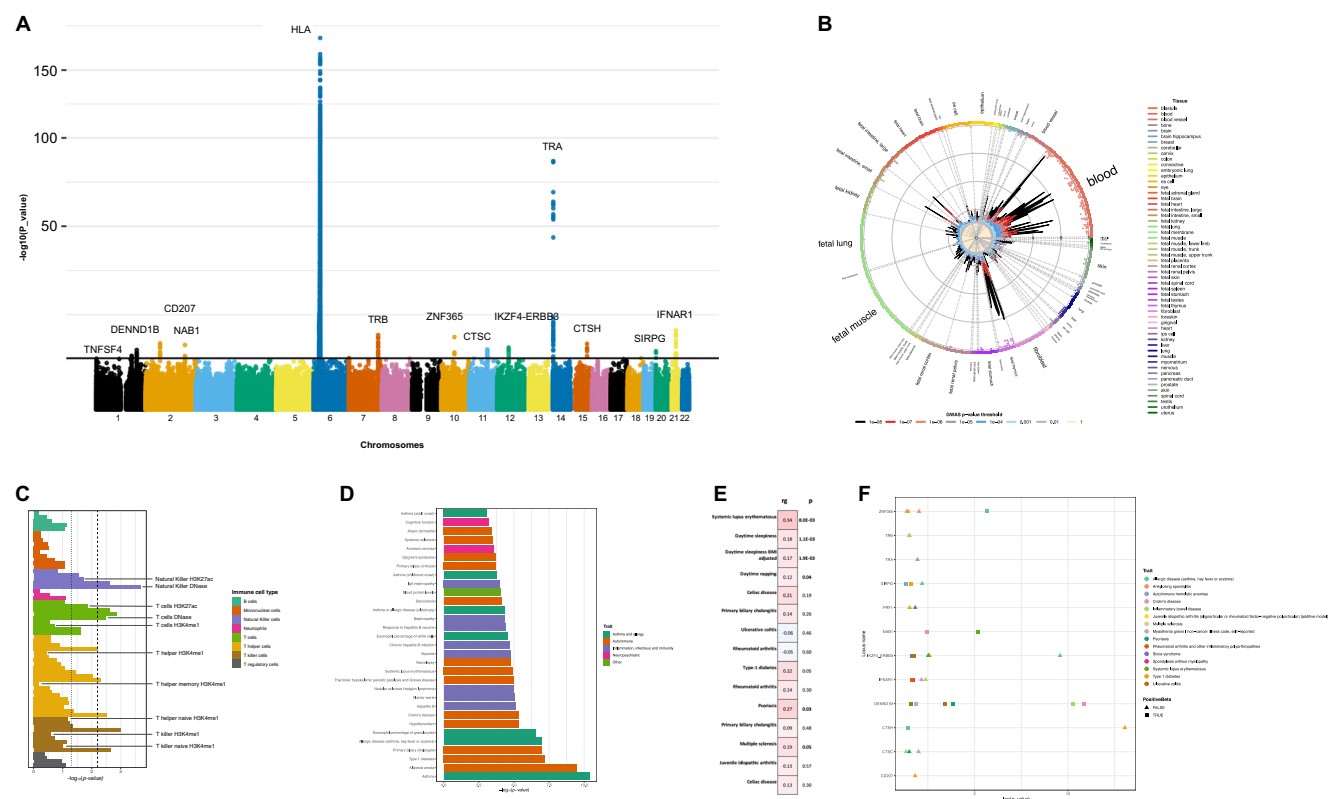

**Fig. 1 | Multi-ethnic genetic analysis of NT1. A** Multi-ethnic analysis conducted in 6073 cases and 84,856 controls reveals significant associations in 13 loci in addition to HLA. The *x*-axis shows genomic location by chromosome and the *y*-axis shows $-\log_{10}$ *P*-values. The red horizontal line indicates the genome-wide significant *P*-value threshold of $5 \times 10^{-8}$. *P*-values larger than $10^{-75}$ were set to $10^{-75}$ (HLA locus has many SNPs with *P*-value < $10^{-216}$). Variants are shared at individual level with known autoimmune traits, with notable exception at variants within the *TRA* and *TRB* loci and variants within *CD207* and *INFAR1* (see Supplementary Data 1). Raw *P*-values are reported using two-sided fixed-effects meta-analysis. Multiple testing correction has been done at genome-wide level so that variants with nominal *P*-value under 5e-8 were considered statistically significant. **B** Associated variants are located on chromosome positions that have active eQTLs in blood samples, as evidenced by an analysis using GARFIELD. Enrichment analysis has been binned by *P*-value threshold and raw two-sided *P*-values are reported. **C** When using stratified LD score regression, association within individual blood cell types implicate NK cells, CD4+ T and CD8+ T cells. Statistically significant enrichment is marked with a line corresponding to enrichment *P*-value = 0.05 (dashed line) and FDR corrected *P*-value = 0.05 (dotted line). Raw two-sided *P*-values are reported and we have show significance also by false-discovery rate of 0.05 (dotted line). **D** Global enrichment is seen with autoimmune traits in general using variants that were genome-wide significant. Raw two-sided *P*-values from hypergeometric test are reported. Fig. 2A, B, E: Raw *P*-values are reported using two-sided fixed-effects meta-analysis. **E** Overall enrichment was seen with MS and SLE using LDSC. **F** PheWAS with narcolepsy risk variants showed association across different autoimmune traits. positive beta is depicted with square and negative with triangle. For details, see methods.

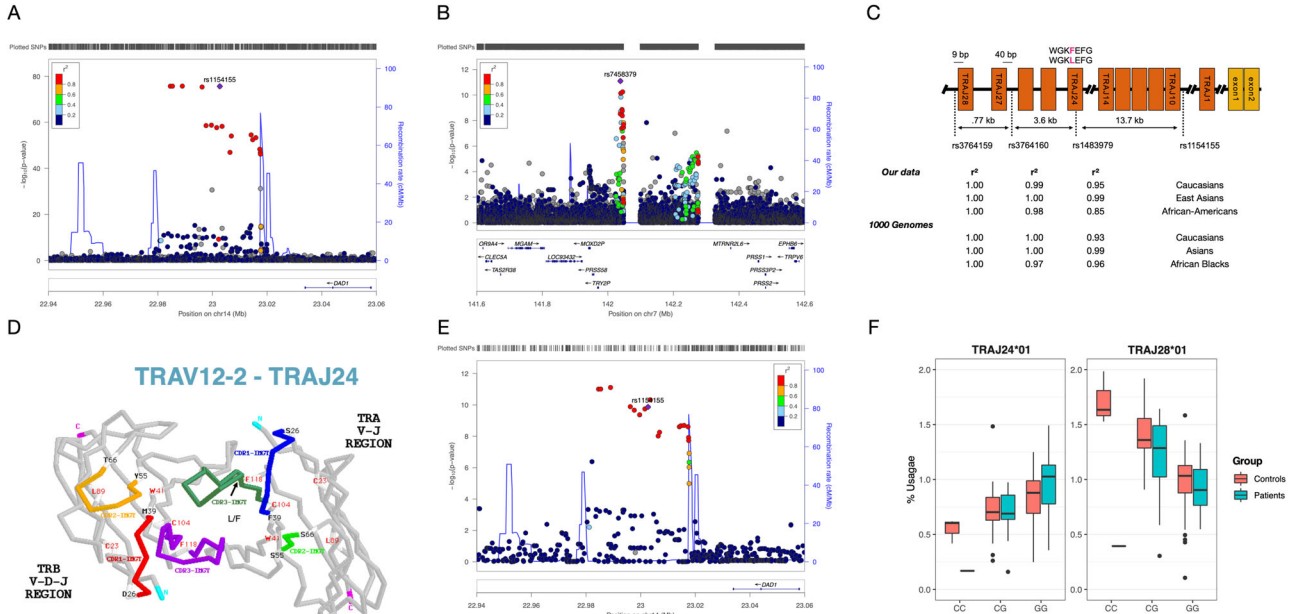

**Fig. 2 | TRA rs1154155 is associated with multiple functional SNPs affecting TRAJ24 and TRAJ28; TRB lead variant affects TRBV 4-2 usage. A** NT1 association with rs1154155 within the T-cell receptor alpha chain (*TRA*) locus. **B** NT1 association with rs7458379 within the T-cell receptor beta chain (*TRB*) locus. **C** The NT1 association within the TRA locus spans a region containing ~30 TRAJ genes that contains 4 SNPs in almost perfect LD (rs1154155, rs1483979, rs3764159, rs3764160) over a 18 kb region. **D** Exemplary TCR receptor structure of a TCR containing TCR J24. rs1483979 encodes a leucine (J24*02) to phenylalanine (J24*01) substitution (F8L) within the J24 segment. The substitution is projected within the Complementary Determining Region (CDR) 3 of the corresponding family of receptors, in an area anticipated to interact directly with the peptide presented by HLA. **E** Usage of

TRAJ28*01 in 895 individuals shows similar association with NT1 lead variant rs1154155, with posterior probability of 0.958 between NT1 and increased TRAJ28 usage. **F** T-cell receptor sequencing in CD4+ T memory cells in 60 NT1 patients and 42 matched controls confirmed the effect of rs1483979 on usage of TRAJ28*01 with similar effect seen in the NT1 cases. It also shows a decreasing effect of rs1483979C on TRAJ24 expression, with a ratio of 24 F/L (associated allele over non-associated allele in ~0.4 in heterozygous subjects, see Supplementary Fig. 5). The center of the boxplot corresponds to the median corresponding to 50th percentile, the box indicates the upper and lower bounds of the interquartile range corresponding to 25th and 75th percentile, and the min and max values correspond to the plus/minus two interquartile ranges.

---

traits including psoriasis, multiple sclerosis and systemic lupus erythematosus in particular ($P < 0.05$, Table S3, Fig. 1E). We computed enrichment of NT1-associated genes with publicly available GWAS data at gene and variant level, identifying overlap with immune and infectious traits such as asthma, type 1 diabetes, primary biliary cholangitis, plantar warts and hepatitis B ($P < 0.001$, Fig. 1D–F and Table S4). Almost all variants identified (*ZNF365, TNFSF4, NAB1, IKZF4-ERBB3, CTSC, DENND1B, SIRPG,* and *PRF1*) are the same or strongly linked with markers associated with other autoimmune diseases (Supplementary Data 1).

In case/control studies, NT1 has been associated with various autoimmune diseases in some[24,25], but not all[26] studies. For example, Chen et al. (2021) found association of narcolepsy with asthma in the general Taiwanese population[27]. To further examine if autoimmune traits are associated with NT1 in epidemiological samples, we explored association between narcolepsy and autoimmune diseases and asthma in 342,499 participants of the FinnGen cohort, retrieving diagnosis of autoimmune diseases, asthma and narcolepsy. In this sample, narcolepsy (157 participants) was associated with psoriasis (OR = 2.29 [1.07–4.90], $P = 0.033$), hypothyroidism (OR = 4.61[2.39–8.93], $P = 5.19*10^{-6}$), rheumatoid diseases (OR = 2.20[1.00–4.82], $P = 0.049$), asthma (OR = 4.58[3.04–6.89], $P = 3.45*10^{-13}$) and "any" autoimmune disease (OR = 2.07[1.50–2.84], $P = 8.31*10^{-6}$). Of note, since DQ0602 is extremely (OR = 0.03) protective against type 1 diabetes[28] and strongly protective (OR = 0.64) against primary biliary cholangitis[29]; no narcolepsy cases had these dual pathologies. Taken together, these findings suggest shared effects between narcolepsy and other autoimmune diseases at both the epidemiological level and at multiple genetic loci, modulated by HLA genotypes.

## Variants involved in antigenic stimulation and infections

The specific polymorphisms in langerin (CD207) we found associated with NT1 have previously been linked to interferon stimulus and influenza uptake by dendritic cells (DC) (Supplementary Data 1). Langerin is a type II transmembrane C-type lectin receptor expressed in Langerhans cells, a specialized type of dendritic cells located exclusively in the respiratory tract and the epidermis, and it recognizes mannose-rich sugars expressed by bacterial, fungal or viral pathogens, including HIV-1 and Influenza-A (Supplementary Data 1).

Our leading variant, rs3815556G, a rare allele, is in complete linkage disequilibrium ($r^2 = 1$) with two coding variants, rs13383830C (N288D) and rs57302492A (K313I) that modulate recognition of bacterial versus viral antigens. The rare Asp-288/Ile-313 haplotype has langerin molecules with enhanced affinity for GlcNAc, present in influenza and other viruses, whereas the other haplotype has higher affinity for high-mannose structures and fucosylated glycans, as well as 6SO4-Gal binding activity. This may potentially allow for protection against a wider range of microorganisms, notably bacterial ones as well (Supplementary Data 1). The NT1-associated variants may thus affect disease predisposition by increasing influenza viral (as opposed to bacterial) uptake and antigen presentation to CD4+ T cells.

In addition to langerin, we identified a regulatory variant near IL10RB-IFNAR1, rs2096464T. The SNP is a strong eQTL for IFNAR1 expression in various tissues in GTEx. IFNAR1 controls dendritic cell responses to viral infections, notably Influenza-A. We therefore examined IFNAR1 expression in DC following H1N1 infection (PR8 delta NS1), finding that the exact NT1 predisposing SNP (rs2096464) is the major eQTL for this effect ($P = 1.92 \times 10^{-25}$, beta = 0.140), as well as for interferon stimulation ($P = 10^{-33}$, beta = 0.215), as it is in perfect

**Table 1 | Association of genetic variants with narcolepsy. Raw P-values are reported using two-sided fixed-effects meta-analysis**

| Closest gene | chr | rsid | Position | Other Allele | Effect Allele | Allele Frequency | Meta-analysis of transethnic cohorts, No vaccination cases | | | Meta-analysis of vaccination cases | | | Meta-analysis of vaccination and non-vaccination cohorts | | | | |
|---|---|---|---|---|---|---|---|---|---|---|---|---|---|---|---|---|---|
| | | | | | | | P-value | beta | SE | P-value Vaccination | beta | se | P-value | beta | se | n cases | n controls |
| TNFSF4 | 1 | rs10158467 | 173131493 | A | G | 0.29 | $6.64 \times 10^{-8}$ | 0.159 | 0.029 | 0.108 | 0.175 | 0.109 | $1.77 \times 10^{-08}$ | 0.16 | 0.028 | 5601 | 86,675 |
| DENND1B | 1 | rs12131588 | 197666111 | G | A | 0.19 | $3.52 \times 10^{-9}$ | 0.17 | 0.029 | 0.191 | 0.164 | 0.126 | $1.46 \times 10^{-09}$ | 0.17 | 0.028 | 5601 | 86,675 |
| CD207 | 2 | rs3815556 | 71059153 | A | G | 0.062 | $3.33 \times 10^{-10}$ | 0.246 | 0.039 | 0.05 | 0.432 | 0.22 | $6.62 \times 10^{-11}$ | 0.251 | 0.039 | 6073 | 87,377 |
| NAB1 | 2 | rs7291718 | 191486081 | C | T | 0.192 | $1.03 \times 10^{-09}$ | -0.205 | 0.034 | 0.039 | -0.288 | 0.139 | $1.38 \times 10^{-10}$ | -0.209 | 0.033 | 5601 | 86,675 |
| TRB | 7 | rs7458379 | 142038166 | C | T | 0.394 | $8.03 \times 10^{-12}$ | 0.155 | 0.023 | 0.019 | 0.306 | 0.13 | $9.40 \times 10^{-13}$ | 0.16 | 0.022 | 5998 | 84,581 |
| ZNF365 | 10 | rs10995245 | 64391375 | G | A | 0.348 | $4.15 \times 10^{-12}$ | 0.148 | 0.021 | 0.302 | 0.106 | 0.103 | $2.62 \times 10^{-12}$ | 0.147 | 0.021 | 6073 | 87,377 |
| CTSC | 11 | rs7112455 | 88051293 | T | A | 0.052 | $5.58 \times 10^{-09}$ | 0.244 | 0.042 | 0.06 | 0.373 | 0.199 | $1.12 \times 10^{-09}$ | 0.249 | 0.041 | 6073 | 87,377 |
| IKZF4-ERBB3 | 12 | rs11171731 | 56443342 | C | T | 0.33 | $5.72 \times 10^{-09}$ | 0.147 | 0.025 | 0.019 | 0.239 | 0.101 | $4.97 \times 10^{-10}$ | 0.152 | 0.024 | 5601 | 86,675 |
| TRA | 14 | rs1154155 | 23002684 | T | G | 0.17 | $2.28 \times 10^{-76}$ | 0.463 | 0.025 | $1.65 \times 10^{-13}$ | 0.85 | 0.115 | $7.31 \times 10^{-86}$ | 0.48 | 0.024 | 6073 | 87,377 |
| CTSH | 15 | rs34593439 | 79234957 | G | A | 0.11 | $2.56 \times 10^{-09}$ | 0.215 | 0.036 | 0.003 | 0.437 | 0.147 | $8.24 \times 10^{-11}$ | 0.227 | 0.035 | 6073 | 87,377 |
| SIRPG | 20 | rs6034239 | 1616137 | G | A | 0.575 | $1.21 \times 10^{-08}$ | 0.123 | 0.022 | 0.056 | 0.215 | 0.113 | $2.61 \times 10^{-09}$ | 0.126 | 0.021 | 5992 | 86,957 |
| IFNAR1 | 21 | rs2096464 | 34686049 | G | T | 0.301 | $4.55 \times 10^{-14}$ | -0.179 | 0.024 | 0.653 | -0.052 | 0.115 | $7.40 \times 10^{-14}$ | -0.174 | 0.023 | 6073 | 87,377 |
| *Novel loci in Z-score-based analysis* | | | | | | | | | | | | | | | | | |
| PRF1 | 10 | rs78325861 | 72378489 | C | G | 0.046 | $2.27 \times 10^{-06}$ | -0.338 | 0.072 | 0.717 | -0.156 | 0.432 | $3.4 \times 10^{-09}$ | -0.333 | 0.071 | 5605 | 85,481 |

Allele frequencies are given for effect allele, position reflects hg19.

LD with the leading variant for the signal (rs6517159, $r^2 = 0.93$, Supplementary Fig. 4). Taken together, these findings suggest that NT1-associated variants may affect disease predisposition by increasing influenza viral (as opposed to bacterial) uptake and antigen presentation to CD4[+] T cells, although additional mechanisms could be involved.

### Fine mapping of multi-loci association in the HLA region
To further fine map the HLA association, we imputed classical HLA class I (HLA-A, HLA-B, HLA-C) and class II (HLA-DRB1, HLA-DQA1, HLA-DQB1, HLA-DPA1 and HLA-DPB1) genes using HIBAG[30] and HLA IMP:02[31] and examined allele associations with NT1. As expected[5,6], the strongest association was with *DQA1\*01:02-DQB1\*06:02* (DQ0602). To delineate additional signals, we performed conditional analysis using stepwise forward regression. We discovered protective associations with *DQA1\*01:01* and *DQA1\*01:03* (OR = 0.30, $P < 10^{-15}$ and OR = 0.30, $P < 10^{-20}$, respectively) and confirmed predisposing effects of *DQB1\*03:01* and *DQA1\*01:02* across ethnic groups, as shown before[5,6,32,33] (OR = 1.23, $P < 0.001$ and OR = 1.47 $P < 1 \times 10^{-6}$, respectively) (Table S5). Finally, controlling for both *DQB1* and *DQA1* effects, a protective association was seen with the *DPB1\*04:02* allele ($P$-value < $10^{-20}$), whereas smaller predisposing effects were found with *DPB1\*05:01* and at HLA class I with *A\*11:01*, *B\*51:01*, *B\*35:01* and *B\*35:03*, and protective association with *A\*03:01* ($P < 0.01$, Table S5). Taken together, these findings confirm and extend results from previous studies[6,33] and highlight independent association of both HLA class I and II alleles with NT1.

### Antigen Presentation and T-cell involvement in NT1
We next examined whether associations with NT1 were enriched genome-wide on specific enhancers using stratified LD score regression (LDSC) on Epigenome Roadmap cell type and ENTEX tissue-specific annotations ($n = 491$ cell and tissue types)[34]. Partitioned heritability by cell type categories was enriched in hematopoietic cell lines (observed $h^2$ at hematopoietic cells = 0.24[0.11], $P = 0.018$) and after partitioning the signal into specific cell subsets, ten cell types showed enrichment with $P < 0.005$. These were either helper or cytotoxic T cells or NK cells (Fig. 1C and Supplementary Data 2). As LDSC does not keep information on the HLA region due to ambiguous linkage disequilibrium, we next examined the contribution of different immune cell types using enrichment analysis with genes close to GWAS significant variants. This analysis further supported the enrichment with CD4+ T cells, but also implicated antigen-presenting cells such as monocytes and dendritic cells ($P$-enrichment < 0.01, Table S6), reflecting, in addition to langerin and IFNAR1, expression of GWA significant, independently associated HLA-associated genes DQB1 and DPB1 (Supplementary Data 1). Together, these results indicate involvement of antigen presentation to CD4[+] and CD8[+] T cells in NT1.

### Risk variants in T-cell receptor loci modulate αβ TCR repertoire
NT1 is the only autoimmune disease with known associations in both HLA and T-cell receptor (TCR) loci (TRA and TRB) (Fig. 2A, B). TCRα and β chains heterodimerize to form biologically functional molecules that recognize peptides presented by HLA. We therefore examined the function of these leading variants by examining effects on T-cell receptor V- or J-gene chain usage using RNA sequencing in 895 individuals[35], as well as in 130 individuals sequenced specifically in both memory CD4[+] and CD8[+] T cells.

As mentioned above, rs1154155 within TRA is entirely linked with multiple SNPs across ethnic groups (Fig. 2C), one of which, rs1483979, substitutes a leucine to a phenylalanine in the CDR3 area of J24, which is predicted to interact with peptides presented by HLA (Fig. 2D; Supplementary Data 5 and 6). This substitution makes it a prime candidate for a functional effect, should an F allele J24*01 CDR3 sequence interact more favorably with an autoantigen than in the presence of

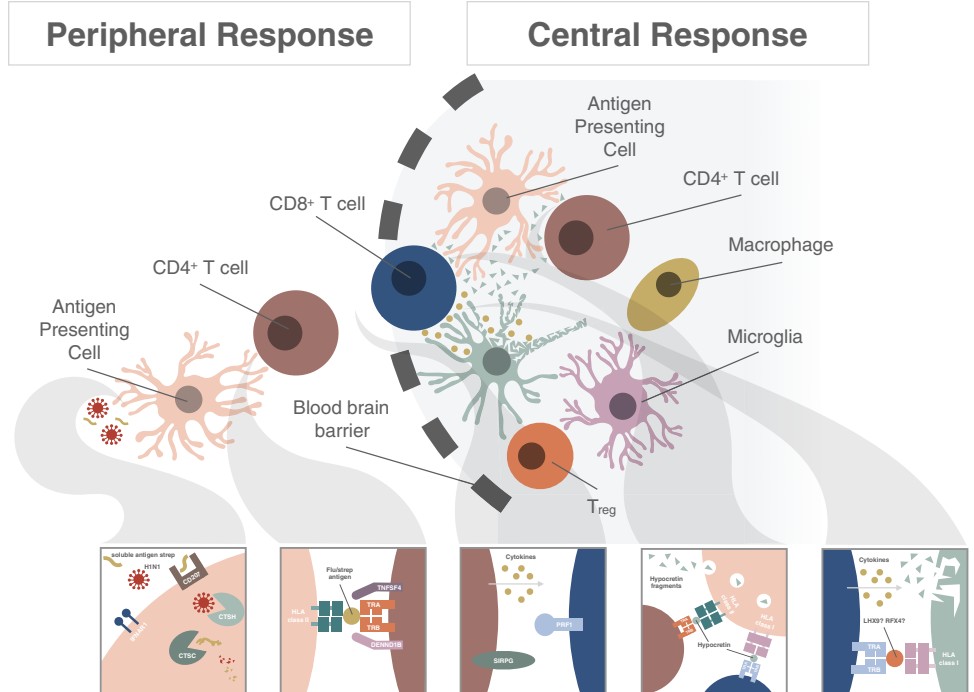

**Fig. 3 | Postulated potential disease mechanisms in autoimmune narcolepsy. (1)** Peripheral response: Influenza virions or vaccine protein debris are ingested by DCs facilitated by CD207; flu proteins are processed by cathepsins CTSH and CTSC for presentation by HLA molecules to specific TCRα-bearing CD4+ cells, initiating an immunological synapse and responses to influenza. Presentation by DC is modulated by IFNAR1 in the context of influenza infection and the type 1 INF response. Cross presentation of influenza antigens processed via the MHC class I pathway in concert with TNFSF4 in DCs is necessary to activate CD8+ cells that mature into cytotoxic lymphocytes (CTLs), initiating cell killing of viron infected cells. Activated Th1 CD4+ cells produce cytokines such as IFNγ and IL-2, which augment cytotoxic activity of CTLs via perforin (PRF1). SIRPG on activated T cells may also promote cell-to-cell adhesion and proliferation in this response. **(2)** CNS autoimmunity: Activated and primed specific CD4+ cells migrate to the CNS, where they interact with microglia and resident DCs via DQ0602 bound to an influenza-mimic autoimmune-epitope (derived from hypocretin cells), initiating a secondary memory response. Hypocretin cell proteins are processed by cathepsins CTSH and CTSC for presentation by DQ0602 to specific TCRα-bearing CD4+ cells, initiating an immunological synapse and autoimmune response. Chain usage for TRAJ24-2, TRAJ28 and TRBV4-2 is associated with NT1 risk and may be crucial for autoantigen recognition. Further, cross presentation by resident DCs and microglial cells activates specific CD8+ cells via MHC class I binding of another HCRT neuron-derived peptide. These primed cytotoxic CD8+ cells then kill HCRT neurons after recognizing MHC class I (such as A*11:01, associated with NT1 independently of DQ0602) bound, cognate HCRT neuron-derived peptide, may be derived from *RFX4 or LHX9*, on hypocretin neurons. SIRPG1 on DCs, microglia or activated T cells may also promote cell-to-cell adhesion and proliferation in this response.

J24*02. Making the matter more complex, however, J24 usage is also modulated by rs1154155, such that the F-associated allele is associated with decreased usage in CD4⁺ cells, as shown by the eQTL plot in Fig. 2F (beta = 0.33, $P < 0.001$). We also computed association in memory CD4+T cells and in CD8+ T cells and observed a consistent effect on TRA-J28 expression specifically ($P = 2.29 \times 10^{-10}$ and $P = 4.08 \times 10^{-10}$, in CD4+ and in CD8+ T cells, respectively). Finally, we confirmed that these effects are *cis* mediated, as the ratio of J24*01 (F) over J24*02 (L) was only 0.4 in heterozygotes, indicating lower allele expression of F-alleles, with similar effects in other T-cell subpopulations (Supplementary Fig. 5).

In addition to J24-specific effects, rs1154155 is also strongly associated with TRA-J28 expression in total RNA sequencing from blood ($P = 1.36 \times 10^{-10}$, beta = −0.212, Fig. 2E) with posterior probability for shared variant pp = 0.958 (see Supplementary Data 3 for all rs1154155 effects) and the findings were also consistent when testing across CD4+ memory cells ($P = 2.29 \times 10$). Interestingly, rs1154155 is entirely linked with rs3764159, a polymorphism located 14 bp upstream of TRAJ-28 within the 12-base pair recombination signal sequence spacer, possibly explaining the J28 usage effect (Fig. 2C). Controlling for the NT1 TCRA association by lead QTL SNPs for J24 and J28 usage abolishes all effects, except for a minor peak at rs72638479, itself a minor eQTL ($r^2 = 0.95$ with TRAV8-6, Supplementary Fig. 6).

Within the TRB region, rs7458379 is an eQTL for increased expression of TRBV4-2. However, based on our functional analysis, rs1108955 has the strongest evidence for increasing TRBV4-2 usage (pp = 0.99, Table S7, Supplementary Data 4). Furthermore, rs1008955 is in partial LD with rs7458379, but tags independent haplotypes at the TRB locus (Supplementary Fig. 7). Both variants are eQTLs for TRBV4-2 expression but reflect independent signals with NT1, such that analysis conditioning for rs7458379 shows remaining association with rs1108955 ($P = 0.00019$), whereas conditioning the association for rs1108955 removed all association at both the TRB locus and with rs7458379 ($P = 0.72$) (Supplementary Fig. 8). Taken together, this indicates that higher expression of TRBV4-2 is related to NT1 and mediated by rs7458379 and rs1108955, with the latter as the potentially causal variant at this locus.

## Discussion

In this study, we explored genetic risk for NT1 and potential disease mechanisms of identified genetic risk factors. The strongest associations were seen within the HLA region. In addition, we confirmed six previously described risk loci (*TRA*, *TRB*, *CTSH*, *IFNAR1*, *ZNF365* and *TNFSF4*) and discovered seven novel associations in *CD207*, *IKZF4-ERBB3*, *NAB1*, *CTSC*, *DENND1B*, *SIRPG*, and *PRF1*.

Individual associations and partitioned heritability enrichment analysis indicate a primary role of the immune system for all loci identified. Most of these loci, often to the exact same SNP, have also been involved in other autoimmune diseases (Supplementary Data 1). These findings, together with the fact that narcolepsy is associated with increased risk of other autoimmune diseases in FinnGen, suggest that NT1 is an autoimmune disease, even if it does not meet all

accepted criteria[36]. Further, most variants identified have effects in antigen-presenting cells (*HLA, CTSH, TNFSF4*), e.g., dendritic cells (*IFNAR1, CD207*), T cells (*TRA, TRB, SIRPG*), T helper cells (*HLA-DQ, HLA-DP*) or cytotoxic T cells/NK cells (*HLA-A, PRF1, NAB1*), sketching a remarkably narrow potential disease pathway (Fig. 3). In addition to epidemiological data, examining genetic factors overcomes disease-associated ascertainment bias in the recruitment of constitutive cohorts. Finally, two loci were implicated at GWA significant level in 245 vaccination-associated NT1 (*TRA* and HLA), while five other loci replicated nominally at $P < 0.05$ (*CD207, NAB1, TRB, IKZF4-ERBB3,* and *CTSH*), with overall strong genetic correlation between sporadic and vaccination-associated cases. This indicates that vaccination-triggered narcolepsy is essentially identical to sporadic narcolepsy.

Unlike what is reported in other autoimmune diseases, however, narcolepsy is strongly associated with TCA and TCB genetic polymorphism that modulate the TCR repertoire in very specific ways. A logical explanation for this observation could be that a TCR-mediated reactivity that involves receptors containing TRAJ24F, TRAJ28 and TRBV4-2 is an important step in the development of narcolepsy, perhaps through TCR recognition of a viral trigger or an autoantigen by CD4[+] or CD8[+] cells. This hypothesis is supported by recent studies suggesting usage of TRAJ24 and TRBV4-2 in the DQ0602 recognition of amidated HCRT, a likely autoantigen, as well as to specific influenza peptides with increased reactivity in narcolepsy[14].

Based on these observations, we propose that NT1 is an autoimmune process where influenza A contributes to risk in the presence of HLA-DQA1\*01:02-DQB1\*06:02 (DQ0602). The involvement of influenza-A may explain why genetic associations found are shared across ethnic groups, as influenza is one of few viruses that act worldwide on a seasonal basis. It also relates to IFNAR1 and that both affect and respond to influenza infections, although other infectious triggers cannot be excluded. Notably, the literature suggests that langerin with Asp-288 and Ile-313 shows no binding to 6SO4-Gal-terminated glycans, increased binding to GlcNAc-terminated structures and overall decreased binding to glycans. This would make langerin more restricted in its ability to bind complex carbohydrates and more able to bind GlcNAc-terminated structures, which overall would favor influenza, but also many other organisms. Similarly, IFNAR1 has been linked to antiviral immunity more generally as well, hence specificity to flu infections cannot be concluded with complete certainty.

The universal genetic association is especially clear for HLA-DQ0602, as it is found with different nearby located HLA-DRB1 alleles: DRB1\*15:01 in individuals of primary European (Europe and USA) and Asian (China, Korea, Japan and India) descent, but DRB1\*15:03 or DRB1\*11:01 in individuals of primary Africa descent[5,6]. The primacy of DQ0602 over DRB1\*15:01 (and thereby DRB5, as LD is complete) is also demonstrated by the fact that the DRB1\*15:01-DQA1\*01:02-DQB1\*06:01 haplotype is not associated with narcolepsy in China and by the fact additional DQ effects are mostly mediated by DQA1 alleles that interact in trans with DQB1\*06:02 (i.e., DQA1\*01:01 and DQB1\*01:03)[33]. Consequently, the association with DRB1\*15:01 was slightly less significant than association with DQ0602. In contrast to NT1, other autoimmune diseases, such as multiple sclerosis and type 1 diabetes, commonly have different HLA associations or disease presentations across countries, resulting in more complex HLA associations. Type 1 diabetes, for example, is well known to be mostly associated with HLA-DRB1\*03:01 and DRB1\*04 and associated DQ alleles in Europe, whereas DRB1\*04:05-specific effects are evident in Japan, where the disease and these DRB1 alleles are rare[37-39].

In this study, a hypomorph of perforin, a gene of critical importance to NK and T-cell cytotoxicity, was protective of NT1. Supporting this, we saw enrichment through tissue-specific partitioned heritability in cytotoxic NK and T cells. Although it is conceivable that NK cells or cytotoxic CD4[+] T cell could be involved, the most likely explanation is involvement of CD8[+] T cell in hypocretin cell killing. Indeed, neurons never express HLA class II, so expression of HLA class I and recognition of hypocretin neuronal antigens would be needed for hypocretin cell pathology to occur. This is also supported by the CTSC association, an enzyme of critical importance to cytotoxic CD8[+] activation of pro-granzymes[40]. Further, Bernard-Valnet et al.[41] used transgenic mice with expression of a neoantigen in HCRT neurons and found that infusion of CD8[+] T-cell targeting the neoantigen were able to cause hypocretin cell destruction, while infusion of neoantigen-specific CD4[+] T-cell alone was insufficient, although CD4[+] T cells migrated closely to the target neurons. Pedersen et al. (2019) also found NT1-associated CD8[+] T-cell targeting intracellular transcription factors such as RFX4 and LXH9, known to be enriched in HCRT cells[15]. Finally, CD8[+] mediation of cell killing has also been suggested by observation of a CD8[+] T-cell infiltrate in a paraneoplastic anti-Ma2 encephalitis case with symptomatic hypocretin cell destruction, although these cases are complex and not associated with DQ0602[42].

In summary, genetic data indicates T-cell autoimmunity in NT1 with genetic overlap of autoimmune traits. A particularity of the disease is involvement of polymorphisms such as in *IFNAR1* and CD207 that regulate antigen-presenting cell responses to infection. Another peculiarity is strong association with TCR polymorphisms, possibly reflecting oligoclonality of T-cell responses. With epidemiological studies indicating seasonality of disease onset[16], and increased incidence following vaccination with Pandemrix® in Europe[17,20], a role of influenza or other infections is likely. CTSH implicates dendritic processing of antigens, perhaps of post-translationally modified HCRT itself[11-15]. Presentation by DQ0602 to CD4[+] T cells ensue, with likely involvement of very few autoantigen epitopes and a restricted number of T-cell receptors, explaining the large effect of these loci. Subsequent cell killing of hypocretin neurons by CD8[+] cells, may be through involvement of other, intracellular autoantigens[43], complete the process. Altogether, this work illustrates how GWAS can identify the involvement of different cell types in a specific condition, thus ultimately providing further insight to the possible pathophysiological mechanisms underlying disease onset.

## Methods

### Study approval

This study has been reviewed and approved by the Stanford University Institutional Review Board (IRB) on Medical Human Subjects (Protocol # 14325, genetic and blood markers in narcolepsy and hypersomnia, Registration # IRB00005136) and by respective IRB panels in each country providing samples for the study. Informed consent was obtained from each participant in accordance with governing institutions. Patients and control subjects in FinnGen provided informed consent for biobank research, based on the Finnish Biobank Act. Alternatively, separate research cohorts, collected prior the Finnish Biobank Act came into effect (in September 2013) and start of FinnGen (August 2017), were collected based on study-specific consents and later transferred to the Finnish biobanks after approval by Fimea, the National Supervisory Authority for Welfare and Health. Recruitment protocols followed the biobank protocols approved by Fimea. The Coordinating Ethics Committee of the Hospital District of Helsinki and Uusimaa (HUS) approved the FinnGen study protocol Nr HUS/990/2017. The FinnGen study is approved by Finnish Institute for Health and Welfare (THL), approval number THL/2031/6.02.00/2017, amendments THL/1101/5.05.00/2017, THL/341/6.02.00/2018, THL/2222/6.02.00/2018, THL/283/6.02.00/2019, THL/1721/5.05.00/2019, Digital and population data service agency VRK43431/2017-3, VRK/6909/2018-3, VRK/4415/2019-3 the Social Insurance Institution (KELA) KELA 58/522/2017, KELA 131/522/2018, KELA 70/522/2019, KELA 98/522/2019, and Statistics Finland TK-53-1041-17. The Biobank Access Decisions for FinnGen samples and data utilized in FinnGen Data Freeze 5 include: THL Biobank BB2017_55, BB2017_111, BB2018_19, BB_2018_34, BB_2018_67, BB2018_71, BB2019_7, BB2019_8, BB2019_26,

Finnish Red Cross Blood Service Biobank 7.12.2017, Helsinki Biobank HUS/359/2017, Auria Biobank AB17-5154, Biobank Borealis of Northern Finland_2017_1013, Biobank of Eastern Finland 1186/2018, Finnish Clinical Biobank Tampere MH0004, Central Finland Biobank 1-2017, and Terveystalo Biobank STB 2018001. (for details, see supplementary materials 1.7 FinnGen). We confirm that all methods were carried out in accordance with relevant guidelines and regulations.

## Study subjects

Six-thousand seventy-three unrelated individuals with NT1, some included in prior studies[8,9], and 84,856 ancestry-matched controls were included in the study. In addition, 245 individuals with vaccination-related NT1 and 18,862 controls were recruited in Finland ($N = 76$ cases and 2796 controls), Sweden ($N = 39$ cases and 4894 controls), Norway ($N = 82$ cases and 429 controls) and United Kingdom and Ireland ($N = 48$ cases and 10,743 controls)[44–47]. All cases had documented immunization with Pandemrix®. All cases had narcolepsy with clear-cut cataplexy and were *DQB1*06:02* positive or had narcolepsy with documented low hypocretin-1 in the cerebrospinal fluid. From FinnGen, we used ICD-10 code G47.4 and ICD-9 code 347, thus including individuals who have narcolepsy with or without cataplexy (for details, see Supplementary Materials 1.7 FinnGen).

## Genotyping

Subjects were genotyped with Affymetrix Affy 5.0, Affy 6.0, Affymetrix Axiom CHB1, Affymetrix Axiom EUR, Axiom EAS, Axiom LAT, Axiom AFR, Axiom PMRA and Human Core Exome chip platforms (Table S1). Genotypes were called with Affypipe[48], Affymetrix genotyping console or Genome Studio. Markers with genotyping quality (call rate < 0.95) or deviation from Hardy-Weinberg equilibrium ($P$-value $< 10^{-6}$) were discarded. Samples were checked for relatedness by filtering based on proportion of identity-by-descent using cut off >0.2 in PLINK 1.9 PI_HAT score. One pair of related individuals was removed. If related individuals were a case and a control, cases were retained in the analysis. Three first principal components within each cohort were visualized and outliers removed.

## Imputation

We imputed samples by pre-phasing cases and controls together using SHAPEIT v2.2 and imputed with IMPUTE2 v2.3.2[49,50] and 1000 Genomes phase 1v3 build37 (hg19) in 5 Mb chunks across autosomes. Haplotype reference consortium data was used for the second Stanford collection. For variants having both imputed and genotyped values, the genotyped values were kept, whereas for those individuals where genotype was missing, imputed values were kept.

## Genetic analyses and statistics

Analyses for all data sets were performed at Stanford University except for the Finnish and Swedish vaccination-related cases and European Narcolepsy Network samples, which were analyzed by respective study teams using the same analytical methods. GWAS was first performed in each case control group separately using SNPTEST v.2.5.2[51]. To adjust for cohort-specific population stratification issues, we used linear regression implemented in SNPTEST method score adjusting for the first ten principal components. Standard post imputation quality control was done. Variants with info score <0.5 and minor allele frequency (MAF) < 0.01 were removed from the analysis. Signals specific for one genotyping platform only and variants in each locus with heterogeneity $P$-value $< 10^{-20}$ were removed. We used a fixed-effects model implemented in METAv1.7 with an inverse-variance method based on a fixed-effects model for combining association results[52]. In total, 12,600,187 markers across studies were included in the final case/control meta-analysis. The significance level for statistically significant association was set to genome-wide significance ($P$-value $< 5 \times 10^{-8}$), controlling for multiple testing. Overall, test statistics showed no genomic inflation

(lambda < 1.05). GCTA was used for heritability and gene-based tests[53]. Coloc analysis was done using coloc package in R version 3.4.2 (2017-09-28)[54] and Manhattan and QQ-plots were created with QQman or FUMA. Shared and partitioned heritability was estimated using LD score regression. To compare the main characteristics of the participants, we used a multivariate logistic regression model as implemented in the R glm package. Lastly, we used FUMA and the curated list of GWAS as provided by FUMA to compute gene enrichment analysis for loci that were associated with narcolepsy at genome-wide significant level and to examine the association between narcolepsy loci and previous GWAS[55]. We also computed genetic correlation between autoimmune traits and narcolepsy using LD score regression and estimated PheWAS associations per lead variant in each locus using the Open targets resource https://genetics.opentargets.org/.

## Comparison of vaccination and non-vaccination cases

Vaccination samples were studied separately for GWAS. To compare genetic architecture of narcolepsy cases following vaccination versus other cases, we first examined association of each GWAS significant SNPs of the primary (non-vaccinated) sample in the vaccination sample. In addition, we computed polygenic risk score in non-vaccination narcolepsy cases and vaccination-related narcolepsy using PRSice[56] and looked at overlap.

## Typing and imputation of HLA variants

High-resolution HLA imputation in 4-digit resolution (2-field, amino acid level) for HLA *A, B, C, DRB1, DQA1, DQB1, DPA1* and *DPB1* was performed using HLA*IMP:02 as implemented in Affymetrix HLA or the HIBAG package in R version 3.1.2 (2014-10-31). HIBAG is an HLA imputation tool that uses attribute bootstrap aggregation of several classifiers (SNPs) to select groups of SNPs that predict HLA type and allows for the use of own HLA reference panels[30]. Reference HLA types were used from published imputation models and for individuals of primary Asian and African descent obtained with Sirona sequencing[57] in ethnic-specific populations ($N = 500$ individuals of African descent, $N = 2000$ individuals of European descent and $N = 368$ individuals of Asian descent). Imputation accuracy was further verified by Luminex HLA typing in a subset of samples and accuracy was over 95% for all ethnic groups and common alleles with >5% frequency in population. For all alleles, the accuracies for individuals of European descent were 98% for HLA-A, 97% for HLA-B, 98% for HLA-C, 96% for HLA-DRB1, 100% for HLA-DQA1, 100% for HLA-DQB1, 100% for HLA-DPA1 and 92% for HLA-DPB1. The accuracies for individuals of Asian descent, where allele typing was also available, were 95% for HLA-DRB1, 94% for HLA-DQA1 and 98% for HLA-DQB1.

## Analysis of HLA variants

HLA effects in NT1 were analyzed as described before[6] in 23,410 individuals, including 9789 individuals of primary Asian descent and 13,621 individuals of primary European descent as ancestry-matched cases and controls. Within each ancestry group, HLA alleles were analyzed using additive models and logistic regressions after adjusting for the first 10 population-specific principal components. We identify independent associations using conditional analysis (stepwise forward regression in each cohort). Fixed-effects meta-analysis was used to combine associations using Plink 1.9[58] and R version 3.2.2. We considered alleles sustaining Bonferroni correction for correction of number of alleles with minor allele frequency over 2% ($N = 110$ HLA alleles), thus significance resulting in Bonferroni cut-off $P = 0.00045$.

## Analysis of expression quantitative trait loci (eQTL)

We used tissue-specific summary statistics from the GTEx consortium and from[59] to examine total blood-specific effects of associating variants on gene expression[59,60]. We used Garfield to compute

enrichments using enhancer annotation data from ENCODE provided by the Garfield software[61] and stratified LD score regression to compute tissue-specific enrichment using ENCODE data as provided by the LCSC package[34].

## Expression assessment in monocyte-derived dendritic cells

We examined how the genetic variants modulated T cell and antigen-presenting (dendritic cell and monocyte) gene expression by RNA sequencing and RNA expression. To examine environment-specific triggers for eQTLs, we challenged dendritic cells with an influenza-A infection or stimulated them with interferon. We recruited individuals free from earlier inflammatory disease, autoimmune disease, chronic metabolic disorders or chronic infectious disorders between 18 and 56 years of age (average 29.9), extracted blood mononuclear cells and differentiated into mononuclear dendritic cells, as previously described[62]. We then extracted RNA from the samples using the RNeasy 96 kit (Qiagen, CAT#74182), according to the manufacturer's protocols and sequenced the samples under baseline, influenza infected and interferon beta 1 (IFNB1) stimulation (99 baseline, 250 influenza infected, and 227 IFNB1 stimulated). Five hundred fifty-two pass-filter samples (94 baseline, 243 influenza, and 215 interferon) were sequenced to an average depth of 38 million 76-bp paired-end reads using the Illumina TruSeq kit. We aligned reads to hg19 genome with TopHat, assembled transcriptomes for each sample using StringTie[63] and computed transcript quantities using Kallisto[64]. We merged transcriptomes across the same condition and then across all three conditions and removed redundant isoforms using cuffcompare[65]. We performed QTL mapping using the Matrix eQTL[66] package using an empirically determined number of principal components (PCs) as covariates in each analysis. We tested 0 to 44 PCs (local eQTLs) in increments of two and the number of PCs was chosen to maximize the number of local eQTLs detected. We computed empirical P-values by comparing the nominal P-values with null P-values determined by permuting each gene 1000 times. False-discovery rates were calculated using the qvalue package (https://github.com/StoreyLab/qvalue), as previously described[67].

## T-cell receptor eQTL analysis

For this analysis, we used data from 895 individuals that were originally genotyped and sequenced as part of the Depression Genes and Networks Project reported by[68], identifying short range (cis) SNPs and trans HLA alleles association with TCR V and J usage as described before[35]. Briefly, expression/usage of each T-cell receptor alpha and beta V- and J-gene was calculated relative to total chain expression from peripheral blood RNA-sequencing. We mapped sequencing reads as in Battle et al. (Bowtie254 with Tophat55 default parameters) and counted the number of unique reads that mapped to each V/J/C-TCR/Ig gene with a modified version of HTSeq56, which allows reads to map to a sequence of more than one V/D/J/C-gene. We then removed individuals and genes with low read counts, normalized the reads using log transformation and regressed on technical and biological covariates as described in Battle et al. as well. Finally, we quantile-normalized the residuals to a normal distribution. Pearson correlations were used to test associations between genotypes and V- or J-gene expression.

## Targeted TCR sequencing in NT1 cases and DQ0602-positive controls

In addition to using the data from Battle et al., (2014) we also conducted TCR sequencing in T cells in 60 individuals with NT1 and 60 healthy DQ0602 individuals using RNA from total CD4+ T cells, CD4+ T memory and CD8+ T-cell populations. We used fastqc to infer quality and trimmed low-quality reads. We then performed barcode demultiplexing, after which local blast was used to align and extract CDR3s. Linear regression was fit for TRA usage per genotype dosage adjusting for age and gender, RNA-sequencing lane and case/control status as covariates. We also separately analyzed coding consequences for each TRAJ24 containing productive CDR3 fragment, as one of the most significantly associated SNPs was a coding SNP (rs1483979) changing an amino acid Leucine to Phenylalanine. These "LQF" and "FQF" were extracted, and their frequencies were computed. Ratio of FQF/(LQF + FQF) was further computed across all samples.

## Reporting summary

Further information on research design is available in the Nature Portfolio Reporting Summary linked to this article.

## Data availability

The sumstats generated in this study have been deposited in the Dryad database under https://doi.org/10.5061/dryad.kd51c5b9b.

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

## Acknowledgements

This study was primarily supported by grants from wake-up narcolepsy, Jazz Pharmaceuticals, donations from narcoleptic patients and, peripherally, by a previously funded NIH-23724 grant (EM). We thank GlaxoSmithKline (GSK) for a collaboration providing funds to examine effect of seasonal influenza vaccination and T-cell receptor sequencing in narcolepsy. Work was also supported by Academy of Finland (#309643), Sigrid Juselius Foundation, Finnish Cultural Foundation, Orion Farmos Research Foundation, Instrumentarium Science Foundation, Oskar Öfflund Foundation, Yrjö Jahnsson Foundation and Jalmari and Rauha Ahokas Foundation (HMO). This work was also supported by the United States Department of Defense (DoD) through the National Defense Science & Engineering Graduate Fellowship (NDSEG) program and by Stanford University through the Stanford Graduate Fellowship program. The work has been supported by the National Institute of Mental Health Grant 5RC2MH089916 for the Depression Genes and Networks Project. The Swedish post-pandemrix Narcolepsy genetics, with Tomas Olsson and Ingrid Kockum as PIs, received support from the Swedish medical product agency. Han Fang was supported by the Ministry of Science and Technology (2015CB856405) and NSFC (81420108002,81670087). Selina Yogeshwar received funding from the Einstein Center for Neurosciences and the German-Academic Exchange Service (DAAD). TCR sequencing eQTL data used sample genotype and sequence as part of the depression gene and networks project; NIMH 5RC2MH089916. We want to thank Dr. Mehdi Tafti for identifying overlapping samples from EU-NN and those genotyped at Stanford University. Some of the computing for this project was performed on the Sherlock cluster. We would like to thank Stanford University and the Stanford Research Computing Center for providing computational resources and support that contributed to these research results. We want to acknowledge the participants and investigators of FinnGen study. The FinnGen project is funded by two grants from Business Finland (HUS 4685/31/2016 and UH 4386/31/2016) and the following industry partners: AbbVie Inc., AstraZeneca UK Ltd, Biogen MA Inc., Bristol Myers Squibb (and Celgene Corporation & Celgene International II Sàrl), Genentech Inc., Merck Sharp & Dohme Corp, Pfizer Inc., GlaxoSmithKline Intellectual Property Development Ltd., Sanofi US Services Inc., Maze Therapeutics Inc., Janssen Biotech Inc, Novartis AG, and Boehringer Ingelheim. Following biobanks are acknowledged for delivering biobank samples to FinnGen: Auria Biobank (www.auria.fi/biopankki), THL Biobank (www.thl.fi/biobank), Helsinki Biobank (www.helsinginbiopankki.fi), Biobank Borealis of Northern Finland (https://www.ppshp.fi/Tutkimus-ja-opetus/Biopankki/Pages/Biobank-Borealis-briefly-in-English.aspx), Finnish Clinical Biobank Tampere (www.tays.fi/en-US/Research_and_development/Finnish_Clinical_Biobank_Tampere), Biobank of Eastern Finland (www.ita-suomenbiopankki.fi/en), Central Finland Biobank (www.ksshp.fi/fi-FI/Potilaalle/Biopankki), Finnish Red Cross Blood Service Biobank (www.veripalvelu.fi/verenluovutus/biopankkitoiminta) and Terveystalo Biobank (www.terveystalo.com/fi/Yritystietoa/Terveystalo-Biopankki/Biopankki/). All Finnish Biobanks are members of BBMRI.fi infrastructure (www.bbmri.fi). Finnish Biobank Cooperative-FINBB (https://finbb.fi/) is the coordinator of BBMRI-ERIC operations in Finland. The Finnish biobank data can be accessed through the Fingenious® services (https://site.fingenious.fi/en/) managed by FINBB.

## Author contributions

H.M.O. and E.J.M. designed the study. H.M.O., E.S., L.L., R.P.H, A. Ambati, J.Y., K.T., T.M., N.R., Z.K., and N.S.A. ran experiments or performed analyses, H.M.O., E.S., N.S.A., and S.M.Y. prepared figures and tables, H.M.O., S.M.Y., and E.J.M. wrote and revised the manuscript. All authors (H.M.O., E.S., L.L., N.S.-A., A. Ambati, S.M.Y., R.P.H., O.J., J.F., M.E., G.L., J.Z., F.H., H.Y., X.S.D., J.L., J.Z., S.-C.H., T.W.K., Y.D., L.B., G.J.L., R.F., G.M., J.S., I.A., S.K., M.K.L.B., P.M.T., G.P., F.P., M.M., C.C., S.K.V.d.E., M.L., P.B., T.K., F.J.M-O., R.P.-A., A.B., J. Montplaisir, A.D., Y.-S.H., FinnGen, P.J., S. Nevsimalova, D.K., A.I., S.O., A.W., P.G., K.S., M.H., B.H., A.S., F.M.C., V.M., E.F., M.W., N.E., H.S., P.H., P.E.H., D.R., Z.P., L.F.-S., C.L.B., J. Mathis, R.K., A. Aran, S. Nampoothiri, T.O., I.K., M. Partinen, M. Perola, B.R.K., S.R., J.W., T.M., H.T., S.S.K., M.S., K.T., M.R., J.K.P., N.R., Z.K., R.O.H., J.H., J.Y. and E.J.M) contributed data. All authors reviewed the manuscript.

## Competing interests

The authors declare no competing interests.

## Additional information

Hanna M. Ollila[1,2,3,4], Eilon Sharon[5,6], Ling Lin[1], Nasa Sinnott-Armstrong[5,6], Aditya Ambati[1], Selina M. Yogeshwar[1,7,8], Ryan P. Hillary[1], Otto Jolanki[5], Juliette Faraco[1], Mali Einen[1], Guo Luo[1], Jing Zhang[1], Fang Han[9], Han Yan[9], Xiao Song Dong[9], Jing Li[9], Jun Zhang[10], Seung-Chul Hong[11], Tae Won Kim[11], Yves Dauvilliers[12], Lucie Barateau[12], Gert Jan Lammers[13,14], Rolf Fronczek[13,14], Geert Mayer[15,16], Joan Santamaria[17], Isabelle Arnulf[18], Stine Knudsen-Heier[19], May Kristin Lyamouri Bredahl[19,20], Per Medbøe Thorsby[20], Giuseppe Plazzi[21,22], Fabio Pizza[21,22], Monica Moresco[21,22], Catherine Crowe[23], Stephen K. Van den Eeden[24], Michel Lecendreux[25], Patrice Bourgin[26], Takashi Kanbayashi[27,28], Francisco J. Martínez-Orozco[29], Rosa Peraita-Adrados[30], Antonio Benetó[31], Jacques Montplaisir[32], Alex Desautels[32], Yu-Shu Huang[33], FinnGen*, Poul Jennum[34], Sona Nevsimalova[35], David Kemlink[35], Alex Iranzo[36,37,38], Sebastiaan Overeem[39,40], Aleksandra Wierzbicka[41], Peter Geisler[42], Karel Sonka[35], Makoto Honda[43,44], Birgit Högl[45], Ambra Stefani[45], Fernando Morgadinho Coelho[46], Vilma Mantovani[47], Eva Feketeova[48], Mia Wadelius[49], Niclas Eriksson[49,50], Hans Smedje[51], Pär Hallberg[49], Per Egil Hesla[52], David Rye[53], Zerrin Pelin[54], Luigi Ferini-Strambi[55], Claudio L. Bassetti[56,57], Johannes Mathis[57], Ramin Khatami[57,58], Adi Aran[59], Sheela Nampoothiri[60], Tomas Olsson[61], Ingrid Kockum[61], Markku Partinen[62,63], Markus Perola[64], Birgitte R. Kornum[65], Sina Rueger[66], Juliane Winkelmann[67,68,69], Taku Miyagawa[43,70], Hiromi Toyoda[43], Seik-Soon Khor[70], Mihoko Shimada[70], Katsushi Tokunaga[70], Manuel Rivas[71], Jonathan K. Pritchard[5], Neil Risch[72], Zoltan Kutalik[66,73], Ruth O'Hara[74,75], Joachim Hallmayer[74,75], Chun Jimmie Ye[76] & Emmanuel J. Mignot[1] ✉

[1]Stanford University, Center for Sleep Sciences and Medicine, Department of Psychiatry and Behavioral Sciences, Palo Alto, CA 94304, USA. [2]Institute for Molecular Medicine Finland, HiLIFE, University of Helsinki, Helsinki, Finland. [3]Anesthesia, Critical Care, and Pain Medicine, Massachusetts General Hospital and Harvard Medical School, Boston, MA, USA. [4]Center for Genomic Medicine, Massachusetts General Hospital, Boston, MA, USA. [5]Department of Genetics, Stanford University, Stanford, CA 94305, USA. [6]Department of Biology, Stanford University, Stanford, CA 94305, USA. [7]Department of Neurology, Charité–Universitätsmedizin, 10117 Berlin, Germany. [8]Charité—Universitätsmedizin Berlin, Einstein Center for Neurosciences Berlin, 10117 Berlin, Germany. [9]Division of Sleep Medicine, The Peking University People's Hospital, Beijing, China. [10]Department of Neurology, The Peking University People's Hospital, Beijing, China. [11]Department of Psychiatry, St. Vincent's Hospital, The Catholic University of Korea, Suwon, Korea. [12]Sleep-Wake Disorders Center, National Reference Network for Narcolepsy, Department of Neurology, Gui-de-Chauliac Hospital, CHU Montpellier; Institute for Neurosciences of Montpellier (INM), INSERM, Université Montpellier 1, Montpellier, France. [13]Department of Neurology, Leiden University Medical Center, Leiden, The Netherlands. [14]Stichting Epilepsie Instellingen Nederland (SEIN), Sleep-Wake Centre, Heemstede, The Netherlands. [15]Hephata Klinik, Schimmelpfengstr. 6, 34613 Schwalmstadt, Germany. [16]Philipps Universität Marburg, Baldinger Str., 35043 Marburg, Germany. [17]Neurology Service, Institut de Neurociències Hospital Clínic, University of Barcelona, Barcelona, Spain. [18]Sleep Disorder Unit, Pitié-Salpêtrière Hospital, Assistance Publique—Hopitaux de Paris, 75013 Paris, France. [19]Norwegian Centre of Expertise for Neurodevelopment Disorders and Hypersomnias (NevSom), Department of Rare Disorders, Oslo University Hospital and University of Oslo, Oslo, Norway. [20]Hormone Laboratory, Department of Medical Biochemistry, Oslo University Hospital, Oslo, Norway. [21]Department of Biomedical and Neuromotor Sciences (DIBINEM), University of Bologna, Via Ugo Foscolo 7, 40123 Bologna, Italy. [22]IRCCS Institute of Neurological Sciences, Bologna, Italy. [23]Mater Private Hospital, Dublin 7, Ireland. [24]Division of Research, Kaiser Permanente Northern California, Oakland, CA, USA. [25]Pediatric Sleep Center and National Reference Center for Narcolepsy and Idiopathic Hypersomnia Hospital Robert Debre, Paris, France. [26]Department of Sleep Medicine, Strasbourg University Hospital, Strasbourg University, Strasbourg, France. [27]Department of Neuropsychiatry, Akita University Graduate School of Medicine, Akita, Japan. [28]International Institute for Integrative Sleep Medicine (WPI-IIIS), University of Tsukuba, Tsukuba, Japan. [29]Sleep Unit. Clinical Neurophysiology Service. San Carlos University Hospital. University Complutense of Madrid, Madrid, Spain. [30]Sleep and Epilepsy Unit, Clinical Neurophysiology Service, Gregorio Marañón University General Hospital and Research Institute, University Complutense of Madrid (UCM), Madrid, Spain. [31]Sleep Unit. Medical Center Valencia, Valencia, Spain. [32]Center for Advanced Research in Sleep Medicine, Hôpital du Sacré-Coeur and Department of Neurosciences, University of Montréal, Montréal, QC, Canada. [33]Department of Child Psychiatry and Sleep Center, Chang Gung Memorial Hospital and University, Taoyuan, Taiwan. [34]Danish Center for Sleep Medicine, Department of Clinical Neurophysiology, University of Copenhagen, Glostrup Hospital, Glostrup, Denmark. [35]Department of Neurology and Centre of Clinical Neurosciences, First Faculty of Medicine, Charles University and General University Hosptal, Prague, Czech Republic. [36]Institut d'Investigacions Biomèdiques August Pi i Sunyer (IDIBAPS), Barcelona, Spain. [37]Department of Neurology, Barcelona, Spain. [38]Multidisciplinary Sleep Disorders Unit, Barcelona, Spain. [39]Sleep Medicine Center Kempenhaeghe, P.O. Box 61, 5590 AB Heeze, The Netherlands. [40]Eindhoven University of Technology, Eindhoven, The Netherlands. [41]Department of Clinical Neurophysiology, Institute of Psychiatry and Neurology, Warsaw, Poland. [42]Department of Psychiatry and Psychotherapy, University of Regensburg, Regensburg, Germany. [43]Department of Psychiatry and Behavioral Sciences, Tokyo Metropolitan Institute of Medical Science, Tokyo, Japan. [44]Seiwa Hospital, Neuropsychiatric Research Institute, Tokyo, Japan. [45]Department of Neurology, Medical University Innsbruck (MUI), Innsbruck, Austria. [46]Universidade Federal de São Paulo, Departamento de Psicobiologia, São Paulo, Brazil. [47]Center for Applied Biomedical Research (CRBA), St. Orsola-Malpighi University Hospital, Bologna, Italy. [48]Neurology Department, Medical Faculty of P. J. Safarik University, University Hospital of L. Pasteur Kosice, Kosice, Slovak Republic. [49]Department of Medical Sciences and Science for Life Laboratory, Uppsala University, Uppsala, Sweden. [50]Uppsala Clinical Research Center, Uppsala, Sweden. [51]Division of Child and Adolescent Psychiatry, Karolinska Institutet, Stockholm, Sweden. [52]Coliseum on Majorstua Clinic, Oslo, Norway. [53]Department of Neurology, Emory University School of Medicine, Atlanta, GA, USA. [54]Faculty of Health Sciences, Hasan Kalyoncu University, Gaziantep, Turkey. [55]Sleep Disorders Center, Division of Neuroscience, Ospedale San Raffaele, Università Vita-Salute, Milan, Italy. [56]Neurology Department, EOC, Ospedale Regionale di Lugano, Lugano, Ticino, Switzerland. [57]Department of Neurology, Inselspital, Bern University Hospital, and University of Bern, Bern, Switzerland. [58]Center for Sleep Medicine and Sleep Research, Clinic Barmelweid AG, Barmelweid, Switzerland. [59]Shaare Zedek Medical Center, Jerusalem, Israel. [60]Department of Pediatric Genetics, Amrita Institute of Medical Sciences & Research Centre, Kerala, India. [61]Department of Clinical Neuroscience, Karolinska Institutet, Stockholm, Sweden. [62]Helsinki Sleep Clinic, Vitalmed Research Centre, Helsinki, Finland. [63]Department of Clinical Neurosciences, University of Helsinki, Helsinki, Finland. [64]University of Helsinki, Institute for Molecular Medicine, Finland (FIMM) and Diabetes and Obesity Research Program. University of Tartu, Estonian Genome Center, Tartu, Estonia. [65]Department of Neuroscience, University of Copenhagen, Copenhagen, Denmark. [66]Swiss Institute of Bioinformatics, Lausanne, Switzerland. [67]Institute of Neurogenomics, Helmholtz Zentrum München, German Research Centre for Environmental Health,

Neuherberg, Germany. [68]Munich Cluster for Systems Neurology (SyNergy), Munich, Germany. [69]Neurologische Klinik und Poliklinik, Klinikum rechts der Isar der Technischen Universität München, Munich, Germany. [70]Department of Human Genetics, Graduate School of Medicine, The University of Tokyo, Tokyo, Japan. [71]Department of Biomedical Data Science-Administration, Stanford University, Palo Alto, CA, USA. [72]Dept. Epidemiology and Biostatistics, UCSF, 513 Parnassus Avenue, San Francisco, CA 94117, USA. [73]University Center for Primary Care and Public Health, University of Lausanne, Lausanne, Switzerland, Lausanne 1010, Switzerland. [74]Department of Psychiatry and Behavioral Sciences, Stanford University, Palo Alto, CA, USA. [75]Mental Illness Research Education Clinical Centers (MIRECC), VA Palo Alto, Palo Alto, CA, USA. [76]Department of Epidemiology & Biostatistics, Institute for Human Genetics, University of California San Francisco, San Francisco, CA, USA. ✉e-mail: mignot@stanford.edu

## FinnGen

Thomas Damm Als[77], Adam Ziemann[78], Ali Abbasi[78], Anne Lehtonen[78], Apinya Lertratanakul[78], Bridget Riley-Gillis[78], Fedik Rahimov[78], Howard Jacob[78], Jeffrey Waring[78], Mengzhen Liu[78], Nizar Smaoui[78], Relja Popovic[78], Adam Platt[79], Athena Matakidou[79], Benjamin Challis[79], Dirk Paul[79], Glenda Lassi[79], Ioanna Tachmazidou[79], Antti Hakanen[80], Johanna Schleutker[80], Nina Pitkänen[80], Perttu Terho[80], Petri Virolainen[80], Arto Mannermaa[81], Veli-Matti Kosma[81], Chia-Yen Chen[82], Heiko Runz[82], Sally John[82], Sanni Lahdenperä[82], Stephanie Loomis[82], Susan Eaton[82], George Okafo[83], Heli Salminen-Mankonen[83], Marc Jung[83], Nathan Lawless[83], Zhihao Ding[83], Joseph Maranville[84], Marla Hochfeld[84], Robert Plenge[83], Shameek Biswas[84], Masahiro Kanai[85], Mutaamba Maasha[85], Wei Zhou[85], Outi Tuovila[86], Raimo Pakkanen[86], Jari Laukkanen[87], Teijo Kuopio[87], Kristiina Aittomäki[88], Antti Mäkitie[89], Natalia Pujol[90], Triin Laisk[90], Katriina Aalto-Setälä[91], Johanna Mäkelä[92], Marco Hautalahti[92], Sarah Smith[92], Tom Southerington[92], Eeva Kangasniemi[93], Henna Palin[93], Mika Kähönen[93], Sanna Siltanen[93], Tarja Laitinen[93], Felix Vaura[94], Jaana Suvisaari[94], Teemu Niiranen[94], Veikko Salomaa[94], Jukka Partanen[95], Mikko Arvas[95], Jarmo Ritari[96], Kati Hyvärinen[96], David Choy[97], Edmond Teng[97], Erich Strauss[97], Hao Chen[97], Hubert Chen[97], Jennifer Schutzman[97], Julie Hunkapiller[97], Mark McCarthy[97], Natalie Bowers[97], Rion Pendergrass[97], Tim Lu[97], Audrey Chu[98], Diptee Kulkarni[98], Fanli Xu[98], Joanna Betts[98], John Eicher[98], Jorge Esparza Gordillo[98], Laura Addis[98], Linda McCarthy[98], Rajashree Mishra[98], Janet Kumar[99], Margaret G. Ehm[99], Kirsi Auro[100], David Pulford[101], Anne Pitkäranta[102], Anu Loukola[102], Eero Punkka[102], Malla-Maria Linna[102], Olli Carpén[102], Taneli Raivio[102], Joni A. Turunen[103,104], Tomi P. Mäkelä[105], Aino Salminen[106], Antti Aarnisalo[106], Daniel Gordin[106], David Rice[106], Erkki Isometsä[106], Eveliina Salminen[106], Heikki Joensuu[106], Ilkka Kalliala[106], Johanna Mattson[106], Juha Sinisalo[106], Jukka Koskela[106], Kari Eklund[106], Katariina Hannula-Jouppi[106], Lauri Aaltonen[106], Marja-Riitta Taskinen[106], Martti Färkkilä[106], Minna Raivio[106], Oskari Heikinheimo[106], Paula Kauppi[106], Pekka Nieminen[106], Pentti Tienari[106], Pirkko Pussinen[106], Sampsa Pikkarainen[106], Terhi Ollila[106], Tiinamaija Tuomi[106], Timo Hiltunen[106], Tuomo Meretoja[106], Tuula Salo[106], Ulla Palotie[106], Antti Palomäki[107], Jenni Aittokallio[107], Juha Rinne[107], Kaj Metsärinne[107], Klaus Elenius[107], Laura Pirilä[107], Leena Koulu[107], Markku Voutilainen[107], Riitta Lahesmaa[107], Roosa Kallionpää[107], Sirkku Peltonen[107], Tytti Willberg[107], Ulvi Gursoy[107], Varpu Jokimaa[107], Aarno Palotie[108], Anastasia Kytölä[108], Andrea Ganna[108], Anu Jalanko[108], Aoxing Liu[108], Arto Lehisto[108], Awaisa Ghazal[108], Elina Kilpeläinen[108], Elisabeth Widen[108], Elmo Saarentaus[108], Esa Pitkänen[108], Hanna Ollila[108], Hannele Laivuori[108], Henrike Heyne[108], Huei-Yi Shen[108], Jaakko Kaprio[108], Joel Rämö[108], Juha Karjalainen[108], Juha Mehtonen[108], Jyrki Pitkänen[108], Kalle Pärn[108], Kati Donner[108], Katja Kivinen[108], L. Elisa Lahtela[108], Mari E. Niemi[108], Mari Kaunisto[108], Mart Kals[108], Mary Pat Reeve[108], Mervi Aavikko[108], Nina Mars[108], Oluwaseun Alexander Dada[108], Pietro Della Briotta Parolo[108], Priit Palta[108], Rigbe Weldatsadik[108], Risto Kajanne[108], Rodos Rodosthenous[108], Samuli Ripatti[108], Sanni Ruotsalainen[108], Satu Strausz[108], Shabbeer Hassan[108], Shanmukha Sampath Padmanabhuni[108], Shuang Luo[108], Susanna Lemmelä[108], Taru Tukiainen[108], Timo P. Sipilä[108], Tuomo Kiiskinen[108], Vincent Llorens[108], Mark Daly[108,109], Jiwoo Lee[85,108], Kristin Tsuo[85,108], Mitja Kurki[85,108], Amanda Elliott[85,108,110], Aki Havulinna[94,108], Juulia Partanen[111], Robert Yang[112], Dermot Reilly[113], Alessandro Porello[114], Amy Hart[114], Dawn Waterworth[114], Ekaterina Khramtsova[114], Karen He[114], Meijian Guan[114], Qingqin S. Li[115], Sauli Vuoti[116], Eric Green[117], Robert Graham[117], Sahar Mozaffari[117], Adriana Huertas-Vazquez[118], Andrey Loboda[118], Caroline Fox[118], Fabiana Farias[118], Jae-Hoon Sul[118], Jason Miller[118], Neha Raghavan[118], Simonne Longerich[118], Johannes Kettunen[119], Raisa Serpi[119], Reetta Hinttala[119], Tuomo Mantere[119], Anne Remes[120], Elisa Rahikkala[120], Johanna Huhtakangas[120], Kaisa Tasanen[120], Laura Huilaja[120], Laure Morin-Papunen[120], Maarit Niinimäki[120], Marja Vääräsmäki[120], Outi Uimari[120], Peeter Karihtala[120], Terhi Piltonen[120], Terttu Harju[120], Timo Blomster[120], Vuokko Anttonen[120], Hilkka Soininen[121], Kai Kaarniranta[121], Liisa Suominen[121], Margit Pelkonen[121], Maria Siponen[121], Mikko Kiviniemi[121], Oili Kaipiainen-Seppänen[121], Päivi Auvinen[121], Päivi Mäntylä[121], Reetta Kälviäinen[121], Valtteri Julkunen[121], Chris O'Donnell[122], Ma'en Obeidat[122], Nicole Renaud[122], Debby Ngo[123], Majd Mouded[123], Mike Mendelson[124], Anders Mälarstig[125], Heli Lehtonen[125], Jaakko Parkkinen[125], Kirsi Kalpala[125], Melissa Miller[125], Nan Bing[125], Stefan McDonough[125], Xinli Hu[125], Ying Wu[125], Airi Jussila[126], Annika Auranen[126], Argyro Bizaki-Vallaskangas[126], Hannu Uusitalo[126], Jukka Peltola[126], Jussi Hernesniemi[126], Katri Kaukinen[126], Laura Kotaniemi-Talonen[126], Pia Isomäki[126], Teea Salmi[126], Venla Kurra[126], Kirsi Sipilä[127,128], Auli Toivola[94], Elina Järvensivu[94], Essi Kaiharju[94], Hannele Mattsson[94], Kati Kristiansson[94], Lotta Männikkö[94],

Markku Laukkanen[94], Markus Perola[94], Minna Brunfeldt[94], Päivi Laiho[94], Regis Wong[94], Sami Koskelainen[94], Sini Lähteenmäki[94], Sirpa Soini[94], Teemu Paajanen[94], Terhi Kilpi[94], Tero Hiekkalinna[94], Tuuli Sistonen[94], Clément Chatelain[129], Deepak Raipal[129], Katherine Klinger[129], Samuel Lessard[129], Fredrik Åberg[130], Mikko Hiltunen[131], Sami Heikkinen[131], Hannu Kankaanranta[132,133,134], Tuula Palotie[135], Iiris Hovatta[136], Kimmo Palin[136], Niko Välimäki[136], Sanna Toppila-Salmi[136], Eija Laakkonen[137], Eeva Sliz[138], Heidi Silven[138], Katri Pylkäs[138], Minna Karjalainen[138], Riikka Arffman[138], Susanna Savukoski[138], Jaakko Tyrmi[138,139], Manuel Rivas[140], Harri Siirtola[139], Iida Vähätalo[139], Javier Garcia-Tabuenca[139], Marianna Niemi[139], Mika Helminen[139] & Tiina Luukkaala[139]

[77]Aarhus University, Aarhus, Denmark. [78]Abbvie, Chicago, IL, US. [79]Astra Zeneca, Cambridge, UK. [80]Auria Biobank/University of Turku/Hospital District of Southwest Finland, Turku, Finland. [81]Biobank of Eastern Finland / University of Eastern Finland / Northern Savo Hospital District, Kuopio, Finland. [82]Biogen, Cambridge, MA, USA. [83]Boehringer Ingelheim, Ingelheim am Rhein, Germany. [84]Bristol Myers Squibb, New York, NY, USA. [85]Broad Institute, Cambridge, MA, USA. [86]Business Finland, Helsinki, Finland. [87]Central Finland Biobank/University of Jyväskylä/Central Finland Health Care District, Jyväskylä, Finland. [88]Department of Medical Genetics, Helsinki University Central Hospital, Helsinki, Finland. [89]Department of Otorhinolaryngology—Head and Neck Surgery, University of Helsinki and Helsinki University Hospital, Helsinki, Finland. [90]Estonian biobank, Tartu, Estonia. [91]Faculty of Medicine and Health Technology, Tampere University, Tampere, Finland. [92]FINBB - Finnish biobank cooperative, Tampere, Finland. [93]Finnish Clinical Biobank Tampere/University of Tampere/Pirkanmaa Hospital District, Tampere, Finland. [94]Finnish Institute for Health and Welfare (THL), Helsinki, Finland. [95]Finnish Red Cross Blood Service/Finnish Hematology Registry and Clinical Biobank, Helsinki, Finland. [96]Finnish Red Cross Blood Service, Helsinki, Finland. [97]Genentech, San Francisco, CA, USA. [98]GlaxoSmithKline, Brentford, UK. [99]GlaxoSmithKline, Collegeville, PA, USA. [100]GlaxoSmithKline, Espoo, Finland. [101]GlaxoSmithKline, Stevenage, UK. [102]Helsinki Biobank/Helsinki University and Hospital District of Helsinki and Uusimaa, Helsinki, Finland. [103]Helsinki University Hospital and University of Helsinki, Helsinki, Finland. [104]Eye Genetics Group, Folkhälsan Research Center, Helsinki, Finland. [105]HiLIFE, University of Helsinki, Finland, Finland. [106]Hospital District of Helsinki and Uusimaa, Helsinki, Finland. [107]Hospital District of Southwest Finland, Turku, Finland. [108]Institute for Molecular Medicine Finland (FIMM), HiLIFE, University of Helsinki, Helsinki, Finland. [109]Broad Institute of MIT and Harvard; Massachusetts General Hospital, Boston, MA, USA. [110]Massachusetts General Hospital, Boston, MA, USA. [111]Institute for Molecular Medicine Finland, HiLIFE, University of Helsinki, Helsinki, Finland. [112]Janssen Biotech, Beerse, Belgium. [113]Janssen Research & Development, LLC, Boston, MA, USA. [114]Janssen Research & Development, LLC, Spring House, PA, USA. [115]Janssen Research & Development, LLC, Titusville, NJ 08560, USA. [116]Janssen-Cilag Oy, Espoo, Finland. [117]Maze Therapeutics, San Francisco, CA, USA. [118]Merck, Kenilworth, NJ, USA. [119]Northern Finland Biobank Borealis / University of Oulu / Northern Ostrobothnia Hospital District, Oulu, Finland. [120]Northern Ostrobothnia Hospital District, Oulu, Finland. [121]Northern Savo Hospital District, Kuopio, Finland. [122]Novartis Institutes for BioMedical Research, Cambridge, MA, USA. [123]Novartis, Basel, Switzerland. [124]Novartis, Boston, MA, USA. [125]Pfizer, New York, NY, USA. [126]Pirkanmaa Hospital District, Tampere, Finland. [127]Research Unit of Oral Health Sciences Faculty of Medicine, University of Oulu, Oulu, Finland. [128]Medical Research Center, Oulu, Oulu University Hospital and University of Oulu, Oulu, Finland. [129]Translational Sciences, Sanofi R&D, Framingham, MA, USA. [130]Transplantation and Liver Surgery Clinic, Helsinki University Hospital, Helsinki University, Helsinki, Finland. [131]University of Eastern Finland, Kuopio, Finland. [132]University of Gothenburg, Gothenburg, Sweden. [133]Seinäjoki Central Hospital, Seinäjoki, Finland. [134]Tampere University, Tampere, Finland. [135]University of Helsinki and Hospital District of Helsinki and Uusimaa, Helsinki, Finland. [136]University of Helsinki, Helsinki, Finland. [137]University of Jyväskylä, Jyväskylä, Finland. [138]University of Oulu, Oulu, Finland. [139]University of Tampere, Tampere, Finland. [140]University of Stanford, Stanford, CA, USA. A full list of members and their affiliations appears in the Supplementary Information.

