## [Peer Review File · Nature Communications]

Narcolepsy risk loci outline role of T cell autoimmunity and infectious triggers in narcolepsyREVIEWER COMMENTS

Reviewer #1 (Remarks to the Author):

In this study, Ollila et al. performed a remarkably large GWAS involving >6000 type 1 narcolepsy cases and over 84,000 controls. They confirmed 6 risk alleles associated with type 1 narcolepsy and discovered 7 additional associations outside the HLA region. They fine mapped the newly identified regions and the already described HLA, TRA and TRB loci and made the very interesting observation that several of the risk alleles are missense variants. Partitioning heritability by cell type indicated enrichment for several immune cell types, including CD4 and CD7 T cells and NK lymphocytes. Finally, some functional studies were performed linking the IL10RB-IFNAR1 variant to response of dendritic cells to viruses and TRA and TRB risk variants to TCR J(alpha) and V(beta) usage or expression levels. This is quite a comprehensive study. The study is well designed. The data are certainly novel and important but their presentation could be improved, to say the least. In the absence of demonstrated viral specificity, the interpretation is unnecessarily biased towards the implication of influenza virus infection.

Comments

1/ This reviewer has great difficulty in analyzing and interpreting the supplementary data. Indeed, the supplementary figures are mostly lacking legends and resemble more a Lab book that a figure for a Nature journal. The supplementary tables are basically unreadable with sentences and tables split in the middle and/or displayed on different pages.

2/ The authors report shared risk loci between type 1 narcolepsy and other autoimmune diseases such as type 1 diabetes, primary biliary cirrhosis and alopecia areata. Has co-occurrence of these diseases with narcolepsy been reported at a frequency above that expected by chance?

3/ The experiments related to expression quantitative trait loci reported on page 12 are loosely describe. It is unclear what precise experiments have been performed. Were viruses other than influenza virus tested in this system? Controls for specificity are required.

4/ Do the TRA gene variants associated with type 1 narcolepsy risk impact TCR usage in specific T cell subset, such as conventional vs. regulatory CD4 T cells or subclasses of CD8 T cells?

5/ Figure 1B, of great scientific importance, is almost impossible to read.

6/ The abstract indicates discovery of 7 new associations whereas, on page 10, 8 novel loci are mentioned. Can you clarify?

7/ The multiple typos should be corrected.

Reviewer #2 (Remarks to the Author):

This is an exceedingly difficult manuscript to review. I am inclined to think that many aspects of the work are valid and the resultant findings are quite novel, yielding considerable advance in our understanding of the biology underlying narcolepsy. Unfortunately, the writing is exceedingly poor at many levels, including an exceptionally large number of spelling and grammatical errors, almost random use of genetical nomenclature, awkward and unclear sentences, and unclear and insufficient explanations. As the result, it is hard to know exactly what was done and even to be confident that things were done correctly. The following are just my most major concerns; there are many more.

1. It is well known that Narcolepsy Type 1 (NT1) is an autoimmune disease caused by loss of HCRT neurons. It is also well known that both Influenza-A and Pandemrix immunization constitute apparent environmental triggers. It is also well known that NT1 is genetically associated with HLA-DQA1*01:02-DQB1*06:02 alleles, encoding the HLA heterodimer DQ0602. Given that background of prior knowledge, while there is much in the current manuscript that is novel, the authors repeated statements about the implications of the current work such as, "Based on these observations, we propose that narcolepsy is an autoimmune process triggered primarily by influenza-A in the presence (sic) of HLA-DQA1*01:02-DQB1*06:02 (DQ0602)" (page 17, and other places) seem overblown.

2. Giving the authors considerable benefit of the doubt, I am inclined to believe the work described is probably generally sound and the results probably constitute a real advancement of scientific knowledge about NT1. However, the writing is so poor, at every level, that it is difficult to be confident. At the lowest level, there are a great many typographic, spelling, and grammar errors, all of which would have been picked up by standard Spellcheck and Grammarcheck routines. At a higher level, use of genetic nomenclature seems nearly random, switching between genetical and protein names as if they were equivalent. Such rampant writing errors that would have been picked up by even rudimentary checking raise concern that the authors might similarly not be so careful in checking their scientific data and conclusions. At an even higher level, there are a great many awkward and unclear sentences and paragraphs, making it hard to fully understand exactly what was done and what the authors are trying to say.

3. A major aspect of the manuscript is sharing of loci/variants with other autoimmune diseases. However, the authors' treatment of this subject is quite superficial. The authors state, "Previous studies have shown either a small increase or no increased risk for autoimmune diseases in NT1 patients" (p. 11). Yet, throughout the manuscript they discuss shared autoimmune associations. It is absolutely required for the current manuscript that the authors present a detailed analysis of the prevalence of other autoimmune diseases in their study cohort, both the 6073-case main cohort (separated by major ethnicity) and the 225 vaccination-induced cases. Similarly, it is essential that the authors present a far more detailed analysis of genetic associations in NT-1 versus these other autoimmune diseases than the summary presentation in Supplementary Tables 4 & 5. They must address at least the following four fundamental questions: 1) What are the prevalences of other autoimmune diseases in their NT-1 cases (and in which populations)?; 2) Which of these represent increased prevalences in NT-1 versus other epidemiological studies of these diseases (and in which populations)? This will require explaining how these diseases were assessed in their study cohorts.; 3) What are the shared genetic associations that correlate with these epidemiological associations?; and 4) To what extent do these shared genetic associations account for the shared epidemiological associations (this last may be hard to answer, but it should at least be discussed)?

4) Another topic of considerable interest is the specific biological relationship between "sporadic" NT-1 and NT-1 subsequent to immunization with Pandemrix. It is not apparent why the authors fail to address this, as they have (and even show) the data. Regarding the 225 Pandemrix cases, they state, "we found GWA (sic; should be GWAS for consistency) significant signals with HLA-DQB1*06:02 and TR rs1154155 (Table 1, Supplementary Fig. 3)". How about the other loci that are genetically associated with sporadic NT-1? While almost certainly the other signals' failure to achieve even nominal significance in most cases results from lack of power due to small sample size, in every case the betas are in the same direction and the meta-analysis of sporadic and vaccination cases shows improved p-values (caveat IFNAR1) over those for the sporadic cases alone. That constitutes strong evidence that the genetic (and thus biological) basis for the two "forms" of NT-1 are substantially overlapping. Oddly, this is not discussed in the present manuscript.

4. The authors provide an interesting biological suggestion regarding why "genetic associations found are universal" (meaning across ethnic groups). However, they provide no data to support their assertion that these associations are "universal". The only analysis presented (Table 1) is meta-analysis of the combined transethnic cohort. While the 10 individual cohorts listed in Supplementary Table 1 likely do not have great individual statistical power, it is absolutely required that the authors present the corresponding underlying genetic analyses of all 10 study cohorts in parallel, so that comparison could be performed and the assertion of "universal associations" be critically evaluated.

Reviewer #3 (Remarks to the Author):

In this manuscript, Ollila et al performed a GWAS in a large cross-cohort across different ethnic groups

to shed light on genetic risk factors for narcolepsy. This study included a sample size three times larger than earlier studies and focused on genetic risk variants associated with the immune system. The analysis confirmed previous findings as well as discovered new genetic association to the risk for type-1 narcolepsy. Overall, these data provide further evidence to the notion of narcolepsy as an autoimmune disorder.

Major comments:

1) Although previous influenza infections and Pandemrix vaccination have been associated to an increased risk for narcolepsy as well as rare autoreactive T cells potentially cross-reactive to flu antigens have been described, the mechanisms behind this remain unclear. Despite the defined correlation between these environmental factors and narcolepsy onset, a clear and direct evidence that they are the causal trigger of the disease is still lacking. Therefore, the authors should rephrase all sentences in the text where this is assumed as unquestionable. For instance: " Known disease triggers" line 156; "'swine flu" pandemic, is a well-established trigger for NT1" line 195; " Finally, as both influenza infections.... Pandemrix® triggers narcolepsy" line 216-217; etc.

2) The authors show an association between CD207 and Narcolepsy patients. In particular, they describe an association with the rare Asp-288/Ile-313 258 haplotype in langerin molecules, which enhances its affinity for GlcNAc. According to the authors, these data indicate that these genetic variants "affect disease predisposition by increasing influenza viral uptake and antigen presentation to CD4+ T cells" and that "strong functional connection with Influenza A infection in dendritic cells was found". However, CD207 has also been shown to mediate the uptake of bacterial and fungi as well as GlcNAc is the main component of bacterial and fungal cell wall. The authors should better discuss about this and explain how their findings could be restricted to Flu infection in the risk of narcolepsy. Do they find enrichment of this variant in post-infection or -vaccination narcolepsy patients?

3) On the same line, IFNAR1 is generally associated with antiviral immunity. How could this genetic variant be directly linked only to Flu infections?

4) As previously described, this study describes a strong association with DQA1*01:02~DQB1*06:02 (DQ0602) haplotype. This haplotype is in linkage disequilibrium with HLA-DRB1*15:01 or HLA-DRB5*01:01. Have you checked the association with these alleles in your analysis?

Minor comments:

1) In the paragraph " Variants involved in response to influenza" some references to the literature are missing

2) Line 301: Fig 1C (not B)

3) Many supplementary tables are cut and consequently the content is not easily readable

REVIEWER RESPONSE

Reviewer #1 (Remarks to the Author):

In this study, Ollila et al. performed a remarkably large GWAS involving >6000 type 1 narcolepsy cases and over 84,000 controls. They confirmed 6 risk alleles associated with type 1 narcolepsy and discovered 7 additional associations outside the HLA region. They fine mapped the newly identified regions and the already described HLA, TRA and TRB loci and made the very interesting observation that several of the risk alleles are missense variants. Partitioning heritability by cell type indicated enrichment for several immune cell types, including CD4 and CD8 T cells and NK lymphocytes. Finally, some functional studies were performed linking the IL10RB-IFNAR1 variant to response of dendritic cells to viruses and TRA and TRB risk variants to TCR J(alpha) and V(beta) usage or expression levels. This is quite a comprehensive study. The study is well designed. The data are certainly novel and important, but their presentation could be improved, to say the least. In the absence of demonstrated viral specificity, the interpretation is unnecessarily biased towards the implication of influenza virus infection.

Thank you for this encouraging comment. We greatly improved presentation of the main text, supplementary material, and figures. As also requested by reviewer 2, we were more cautious regarding involvement of influenza in some of these genetic associations.

Comments

1) This reviewer has great difficulty in analyzing and interpreting the supplementary data. Indeed, the supplementary figures are mostly lacking legends and resemble more a Lab book than a figure for a Nature journal. The supplementary tables are basically unreadable with sentences and tables split in the middle and/or displayed on different pages.

R1.1 We apologize for the poor presentation, which we have now improved. We expanded the supplementary materials, modified the supplementary tables and figures, and added additional methodological details. We welcome any additional suggestions you may have.

2) The authors report shared risk loci between type 1 narcolepsy and other autoimmune diseases such as type 1 diabetes, primary biliary cirrhosis, and alopecia areata. Has co-occurrence of these diseases with narcolepsy been reported at a frequency above that expected by chance?

R1.2. This is a very interesting point. Regarding type 1 diabetes, the overlap is predicted to be extremely low as DQ0602 is 99% protective for T1 diabetes (a phenomenon that remains to be explained)(Simmons et al., 2020). Similarly, DQ0602 is also reported to be (OR=0.6) protective in primary biliary cirrhosis (PBC)(Mullarkey et al., 2005). On the other hand, there are several case reports on the comorbidity of narcolepsy and alopecia areata (King et al., 2010; Nigam et al., 2016). Alopecia is primarily associated with DRB1*04:01 (which would reduce a bit the incidence of DQ0602 in these subjects) and multiple other effects in the HLA region, but no protective effects of DQ0602 have been reported on alopecia.

Some studies (Feketeova et al., 2020; Martinez-Orozco et al., 2016), but not all (Barateau et al., 2017) have found increased co-occurrence of all autoimmune disorders in case-control studies, but these studies have serious methodological limitations as ascertainment was likely not systematically done in comparison to controls. Interestingly, a population-based study from Taiwan found a significant association (OR=3.1, $p < 0.001$) between asthma and narcolepsy (Chen et al., 2021). In our study, narcolepsy strongly shares genetic architecture with asthma and atopic disorders. Further, HLA effects in asthma are complex and do not particularly involve DQB1*06:02, thus overlap is to be expected, unlike with type 1 diabetes.

As this question was also raised by reviewer 2, we looked at these associations in the UK Biobank and FinnGenn, confirming increased prevalence of autoimmune diseases and asthma in narcolepsy versus matched controls, information that was also added to this manuscript. While the number of individuals with narcolepsy in the UK Biobank was less than 50, we identified 157 individuals in FinnGen with access to electronic health record data based on ICD9 and ICD10 diagnosis (www.finnngen.fi). The total sample size was 342,499 and the prevalence of narcolepsy type 1 in this sample was 0.00046, which is remarkably like what has been reported before (Kornum et al., 2017). Out of the 157 individuals with narcolepsy, 67 had at least one other autoimmune disease (excluding narcolepsy) (OR = 2.07 logSE = 0.163, $P = 8.31 \times 10^{-6}$), thus suggesting enrichment of comorbid autoimmune disorders with narcolepsy. To complete this data, we computed epidemiological association statistics in this data set with individual, specific autoimmune traits and with asthma: five immune traits had a high enough frequency in narcolepsy individuals to perform meaningful statistical analysis (over five individuals with disease), revealing that the prevalence of psoriasis, rheumatoid arthritis, hypothyroidism and asthma were higher in the narcolepsy population.

The paper was edited as follows:

„To confirm that autoimmune traits are associated with NT1 in epidemiological samples, we explored association between narcolepsy and autoimmune diseases and asthma in 342,499 participants of the FinnGen cohort, retrieving diagnosis of autoimmune diseases, asthma and narcolepsy. In this sample, narcolepsy (157 participants) was associated with psoriasis (OR = 2.29, logSE = 0.39), hypothyroidism (OR = 4.61, logSE = 0.34), rheumatoid diseases (OR = 2.20, logSE = 0.40), asthma (OR = 4.57, logSE = 0.21) and “any” autoimmune disease (OR = 2.07, logSE = 0.21). Of note, since DQ0602 is extremely (OR=0.03) protective against type 1 diabetes²⁸ and strongly protective (OR=0.64) against primary biliary cholangitis²⁹, no narcolepsy cases had these dual pathologies. Taken together, these findings suggest shared effects between NT1 and other autoimmune diseases at both the epidemiological level and at multiple genetic loci, modulated by HLA genotypes.”

3) The experiments related to expression quantitative trait loci reported on page 12 are loosely described. It is unclear what precise experiments have been performed. Were viruses other than influenza virus tested in this system? Controls for specificity are required.

R1.3. We apologize and have updated the method section to describe the stimulation and baseline eQTL analyses performed in dendritic cells. Conditions included baseline, H1N1 influenza infection and interferon beta 1 stimulation. All these conditions were assessed in subjects with and without the IFNAR1 variant. This setting allowed us to examine influenza-specific vs. interferon and influenza shared associations. We did not test other pathogens. This is now described in the method section, and the publication from which the data was generated, Ye et al. (Ye et al., 2018), cited.

4) Do the TRA gene variants associated with type 1 narcolepsy risk impact TCR usage in specific T cell subset, such as conventional vs. regulatory CD4 T cells or subclasses of CD8 T cells?

R1.4. To answer this question, we computed chain usage for two different T cell subsets: CD4 memory cells and CD8 T cells. Naïve T cells and Tregs have not been tested due to lack of material. Even across these two different types of T cells (memory CD4 cells vs. CD8), lead variant rs1154155, and the associated missense variant rs1483979, had a similar effect size on TRA-J28 expression (CD4 memory cells: rs1483979 beta = 1.212, $P = 4.2 \times 10^{-10}$, and in CD8 cells, beta = 1.211, $P = 5.23 \times 10^{-9}$). Similarly, we observed consistent effects with rs1154155 (CD4: rs1154155 beta = 1.16, $P = 2.29 \times 10^{-10}$ and CD8: beta = 1.221, $P = 4.08 \times 10^{-10}$). We added supplementary tables 12 and 13 to describe the association in these cell types for rs1154155 (Supplementary table 12), and rs1483979 (Supplementary table 13) for all chains tested. In addition, we outline these results in the main text as follows:

“We also computed association in memory CD4+ T cells and in CD8+ T cells and observed a consistent effect on TRA-J28 expression specifically ($p = 2.29 \times 10^{-10}$ and $p = 4.08 \times 10^{-10}$, in CD4+ and in CD8+ T cells, respectively).”

5) Figure 1B, of great scientific importance, is almost impossible to read.

R1.5 To ensure clarity and readability, Figure 1B was re-formatted. Thank you very much for pointing this out.

6) The abstract indicates discovery of 7 new associations whereas, on page 10, 8 novel loci are mentioned. Can you clarify?

R1.6 Only 7 new loci are reported. Thank you for catching this error!

7) The multiple typos should be corrected.

R1.7 This comment was made also by other reviewers, and we therefore have carefully reviewed the text and requested a professional English speaker to correct the language.

Reviewer #2 (Remarks to the Author):

This is an exceedingly difficult manuscript to review. I am inclined to think that many aspects of the work are valid, and the resultant findings are quite novel, yielding considerable advance in our understanding of the biology underlying narcolepsy. Unfortunately, the writing is exceedingly poor at many levels, including an exceptionally large number of spelling and grammatical errors, almost random use

of genetical nomenclature, awkward and unclear sentences, and unclear and insufficient explanations. As the result, it is hard to know exactly what was done and even to be confident that things were done correctly. The following are just my most major concerns; there are many more.

1) It is well known that Narcolepsy Type 1 (NT1) is an autoimmune disease caused by loss of HCRT neurons. It is also well known that both Influenza-A and Pandemrix immunization constitute apparent environmental triggers. It is also well known that NT1 is genetically associated with HLA-DQA1*01:02-DQB1*06:02 alleles, encoding the HLA heterodimer DQ0602. Given that background of prior knowledge, while there is much in the current manuscript that is novel, the authors repeated statements about the implications of the current work such as, "Based on these observations, we propose that narcolepsy is an autoimmune process triggered primarily by influenza-A in the presence of HLA-DQA1*01:02-DQB1*06:02 (DQ0602)" (page 17, and other places) seem overblown.

R2.1. These sentences have been rewritten and toned down. We also revised the manuscript to outline more clearly what is novel 1) doubling the size of known associated loci; 2) further analysis of the functional pathways involved; 3) functional characterization of TCR and INFAR1 polymorphisms; 4) studies of vaccination associated narcolepsy; 5) epidemiological association of narcolepsy with other autoimmune diseases in FinnGenn (an addition to the manuscript). All these novel findings have important implications for our understanding of narcolepsy pathophysiology.

2) Giving the authors considerable benefit of the doubt, I am inclined to believe the work described is probably generally sound and the results probably constitute a real advancement of scientific knowledge about NT1. However, the writing is so poor, at every level, that it is difficult to be confident. At the lowest level, there are a great many typographic, spelling, and grammar errors, all of which would have been picked up by standard Spellcheck and Grammarcheck routines. At a higher level, use of genetic nomenclature seems nearly random, switching between genetical and protein names as if they were equivalent. Such rampant writing errors that would have been picked up by even rudimentary checking raise concern that the authors might similarly not be so careful in checking their scientific data and conclusions. At an even higher level, there are a great many awkward and unclear sentences and paragraphs, making it hard to fully understand exactly what was done and what the authors are trying to say.

R2.2. We apologize and have largely rewritten the manuscript. We also asked a professional English editor to correct the language. Furthermore, we now consistently used italics when referring to a specific gene and no italics when referring to expression.

3) A major aspect of the manuscript is sharing of loci/variants with other autoimmune diseases. However, the authors' treatment of this subject is quite superficial. The authors state, "Previous studies have shown either a small increase or no increased risk for autoimmune diseases in NT1 patients" (p. 11). Yet, throughout the manuscript they discuss shared autoimmune associations. It is absolutely required for the current manuscript that the authors present a detailed analysis of the prevalence of other autoimmune diseases in their study cohort, both

the 6073-case main cohort (separated by major ethnicity) and the 225 vaccination-induced cases.

Similarly, it is essential that the authors present a far more detailed analysis of genetic associations in NT-1 versus these other autoimmune diseases than the summary presentation in Supplementary Tables 4 & 5. They must address at least the following four fundamental questions: 1) What are the prevalences of other autoimmune diseases in their NT-1 cases (and in which populations)?; 2) Which of these represent increased prevalence in NT-1 versus other epidemiological studies of these diseases (and in which populations)? This will require explaining how these diseases were assessed in their study cohorts.; 3) What are the shared genetic associations that correlate with these epidemiological associations? and 4) To what extent do these shared genetic associations account for the shared epidemiological associations (this last may be hard to answer, but it should at least be discussed)?

R2.3. This is an important question, which was also raised by reviewer 1. Unfortunately, we do not have access to individual cohort level data on associated autoimmune diseases from participating sites. It would be materially unfeasible to gather this data across all these countries at this stage, not to mention that since data on other autoimmune diseases has not been systematically ascertained, it would be unreliable.

The same problem also applies to vaccination-triggered cases, but for a different reason: Most of the vaccination-triggered cases have been young children since the disease is faster to identify (more abrupt and dramatic onset). Further, risk was more increased in children (perhaps due to the fact it was the first flu antigenic exposure in many children and that the adjuvanted vaccine very strongly stimulated CD4 responses). As occurrence of various autoimmune diseases is age dependent (for example, type 1 diabetes occurs at a similar age range as narcolepsy, lupus often affects women of age 30, anti-Lgl1 encephalitis affects males of age 60, etc.), ascertainment at narcolepsy onset in post-vaccination cases would unlikely be representative of true risk of comorbidity.

To remedy the situation, we are now better citing the literature, which in general supports increased prevalence of all autoimmune diseases and asthma in narcolepsy cases. We also note this is confounded for individual diseases by the fact HLA effects are dominant and can have overwhelming effects in some cases (see response to reviewer 1). More importantly, we also obtained access to population cohort data from Finland, where clinical diagnostic history was based on ICD9 and ICD10 codes available from inpatient and outpatient registries in 342,499 study participants, of which 157 participants had type 1 narcolepsy. Using this data from adults, we looked at prevalence of autoimmune diseases, asthma and narcolepsy answering the following questions:

What are the prevalences of other autoimmune diseases in their NT1 cases (and in which populations)?

Of 342,499 FinnGen study participants, 157 had narcolepsy. In these patients, rheumatoid arthritis and psoriasis were present in 4.5%, Hypothyroidism 14% and Asthma 25%. Many autoimmune traits had too low a case number to be analyzed on their own. The presence of any autoimmune disease was observed in 43% of narcolepsy cases, a strikingly high figure.

Which of these represent increased prevalences in NT-1 versus other epidemiological studies of these diseases (and in which populations)?

There are very few studies that have looked at the prevalence of autoimmune diseases in narcolepsy. As mentioned above, one of these studies (Chen et al., 2021) found that asthma was associated with narcolepsy (OR = 3.81) in a Taiwanese sample using Taiwan's National Health Insurance Research Database. Other studies have found increased, (Feketeova et al., 2020; Martinez-Orozco et al., 2016), or similar (Barateau et al., 2017) prevalence of any autoimmune diseases versus controls. Worth noting, both latter three studies are case-only studies and differ from population-based analyses (see answers to reviewer 1), such as the one we perform in the FinnGen dataset.

The results provided through analysis of FinnGen data support that autoimmune diseases are enriched in individuals with narcolepsy (OR = 2.07 logSE = 0.163, P = 8.31×10^{-6}). Like the study from Taiwan, association was particularly noteworthy between asthma and narcolepsy (OR = 4.57 logSE = 0.21, P = 3.43×10^{-13}). All in all, both the literature and the additional epidemiological data therefore support the shared genetic autoimmune traits identified in our current study.

What are the shared genetic associations that correlate with these epidemiological associations?

Asthma was systematically associated with narcolepsy in the epidemiological cohort, both by genetic correlation by LDSC and at the level of individual variants. We did not find shared genetic associations with Lupus or Multiple Sclerosis and the case numbers in individuals with narcolepsy were less than six individuals.

To what extent do these shared genetic associations account for the shared epidemiological associations?

Overall, genetic correlations between narcolepsy and autoimmune traits were modest. This suggests that while many underlying genes are involved across autoimmune diseases, there are also unique, disease specific alleles and environmental mechanisms that contribute to the development of narcolepsy versus other diseases. As discussed in the answer to reviewer 1, susceptibility to many individual autoimmune diseases involves a predominant HLA background that often contributes the most to disease predisposition and specificity. Taking the example of type 1 diabetes, DQ0602, a factor 97% associated with narcolepsy, is 99% protective against diabetes type 1. In this case, even if there is shared heritability at many loci, the different HLA alleles are making it impossible to see these two conditions in the same individuals. This limits interest in comparing co-occurrence for specific autoimmune diseases. This is now also briefly discussed in the manuscript as follows:

"Of note, since DQ0602 is extremely (OR=0.03) protective against type 1 diabetes and protective (OR=0.64) against primary biliary cholangitis, no narcolepsy cases had these dual pathologies."

Overall, we have edited the manuscript to include our epidemiological findings and discuss the association between asthma, overall autoimmunity and narcolepsy as comorbidities in the discussion section more extensively. We believe this addition greatly improves the manuscript, and we thank this reviewer for asking us to look into it.

4) Another topic of considerable interest is the specific biological relationship between "sporadic" NT-1 and NT-1 after immunization with Pandemrix. It is not apparent why the authors fail to address this, as they have (and even show) the data. Regarding the 225 Pandemrix cases, they state, "we found GWA (sic; should

be GWAS for consistency) significant signals with HLA-DQB1*06:02 and TR rs1154155 (Table 1, Supplementary Fig. 3)". How about the other loci that are genetically associated with sporadic NT-1? While almost certainly the other signals' failure to achieve even nominal significance in most cases results from lack of power due to small sample size, in every case the betas are in the same direction and the meta-analysis of sporadic and vaccination cases shows improved p-values (caveat IFNAR1) over those for the sporadic cases alone. That constitutes strong evidence that the genetic (and thus biological) basis for the two "forms" of NT-1 are substantially overlapping. Oddly, this is not discussed in the present manuscript.

R2.4 This is a valid point and we have elaborated discussion of this in the manuscript in two different ways: First, we computed polygenic risk scores (PRS) from the non-vaccinated GWAS sample. Second, we used this to plot the variance explained by this PRS when used to predict associations in the post vaccination cohort, the y-axis being the % variance explained by the PRS, while the x-axis represents the p-value thresholds (Supplementary figure 3C). As can be seen, significance is very high. Second, after noting in the text which loci of the primary GWAS are nominally significant in the vaccination cohort (from Table 1), we added the following to the main text:

"The lack of association of other loci is likely due to the small number of individuals with vaccination-related narcolepsy [N=245] and consequently, reduced power to observe associations."

5) The authors provide an interesting biological suggestion regarding why "genetic associations found are universal" (meaning across ethnic groups). However, they provide no data to support their assertion that these associations are "universal". The only analysis presented (Table 1) is meta-analysis of the combined transethnic cohort. While the 10 individual cohorts listed in Supplementary Table 1 likely do not have great individual statistical power, it absolutely required that the authors present the corresponding underlying genetic analyses of all 10 study cohorts in parallel, so that comparison could be performed and the assertion of "universal associations" be critically evaluated.

R2.5. We have updated the manuscript and supplementary files (specifically, please refer to Supplementary Table 2) to include association statistics in each cohort. In addition, we provide meta-analysis results per ethnic group. We also provide heterogeneity p-value within each ethnic group to evaluate heterogeneity to allow examining between-cohort and between ethnic group associations. Although the associations are generally found across all ethnic groups (although in the small African American cohort, it is often not visible), we observed significant heterogeneity with *TRA*, *DENND1B* and with *SIRPG* and have updated the manuscript as follows:

*"We observed that most associations were shared across all ethnic groups. Significance between-cohort heterogeneity was observed with *TRA*, *SIRPG* and *DENND1B* (Table S2)."*

Reviewer #3 (Remarks to the Author):

In this manuscript, Ollila et al performed a GWAS in a large cross-cohort across different ethnic groups to shed light on genetic risk factors for narcolepsy. This

study included a sample size three times larger than earlier studies and focused on genetic risk variants associated with the immune system. The analysis confirmed previous findings as well as discovered new genetic association to the risk for type 1 narcolepsy. Overall, these data provide further evidence to the notion of narcolepsy as an autoimmune disorder.

Major comments:

1) Although previous influenza infections and Pandemrix vaccination have been associated to an increased risk for narcolepsy as well as rare autoreactive T cells potentially cross-reactive to flu antigens have been described, the mechanisms behind this remain unclear. Despite the defined correlation between these environmental factors and narcolepsy onset, a clear and direct evidence that they are the causal trigger of the disease is still lacking. Therefore, the authors should rephrase all sentences in the text where this is assumed as unquestionable. For instance: " Known disease triggers" line 156; "'swine flu" pandemic, is a well-established trigger for NT1" line 195; " Finally, as both influenza infections.... Pandemrix® triggers narcolepsy" line 216-217; etc.

R3.1 As requested by all reviewers, this was modified and the according statements formulated more tentatively. We believe that the connection with influenza is strong based on biology, epidemiology and vaccination data, but this reviewer is correct in noting that it is impossible to exclude an effect of other infections. For example, it remains to be explained why in many cases increased Streptococcus Pyogenes infection has also been reported (coinfections such as influenza and streptococcus frequently occur, however). Further, it is important to note that even after Pandemrix®, only one child in 16,000 (thus roughly only 1 per 4,000 HLA-DQ0602 positive vaccinated children) developed narcolepsy. Keeping this in mind, it is easy to speculate that other factors and maybe other infections are required. It is also notable that only 25% of monozygotic twins are concordant.

2) The authors show an association between CD207 and Narcolepsy patients. They describe an association with the rare Asp-288/Ile-313 haplotype in langerin molecules, which enhances its affinity for GlcNAc. According to the authors, these data indicate that these genetic variants "affect disease predisposition by increasing influenza viral uptake and antigen presentation to CD4+ T cells" and that "strong functional connection with Influenza A infection in dendritic cells was found". However, CD207 has also been shown to mediate the uptake of bacterial and fungi as well as GlcNAc is the main component of bacterial and fungal cell wall. The authors should better discuss about this and explain how their findings could be restricted to Flu infection in the risk of narcolepsy. Do they find enrichment of this variant in post-infection or -vaccination narcolepsy patients?

R3.2. This reviewer is correct in pointing out that GLcNac is not exclusive to influenza and exists in some bacteria and fungus. What was shown in the literature is that Langerin with Asp-288 and Ile-313 shows no binding to 6SO4-Gal-terminated glycans, increased binding to GlcNAc-terminated structures and overall decreased binding to glycans. This would make langerin more restricted in its ability to bind complex carbohydrates and more able to bind GlcNAc-terminated structures, which overall would favor influenza, but also many other organisms. It is also interesting to note that as for INFAR1, the CD207

polymorphism has not been found in other autoimmune disorder GWASes. We have added this to the discussion as follows:

"Notably, the literature suggests that langerin with Asp-288 and Ile-313 shows no binding to 6SO4-Gal-terminated glycans, increased binding to GlcNAc-terminated structures and overall decreased binding to glycans. This would make langerin more restricted in its ability to bind complex carbohydrates and more able to bind GlcNAc-terminated structures, which overall would favor influenza, but also many other organisms."

Furthermore, in table 1 we show association statistics for this variant in post-vaccination cases with p-values below 0.05. This was not a surprise to us, as when we analyzed glycosylation amounts in influenza vaccine preparations such as Pandemrix, we found extensive glycosylation of HA and other antigens (Jacob et al., 2015) in vaccine antigens, so that this uptake mechanism would apply to vaccines (presumably, in epidermis Langerhans cells) as much as to the live virus (presumably, in upper airway and lung Langerhans cells, the only other known location with the skin). Although compatible with the influenza hypothesis, we report this to the benefit of this reviewer; we did not discuss this in the manuscript as we felt it was too far-fetched.

3) On the same line, IFNAR1 is generally associated with antiviral immunity. How could this genetic variant be directly linked only to flu infections?

R3.3 As part of editing the causality claims, we have also modified the influenza connection. It is possible that the genetic variants from *IFNAR1* also contribute to other infections, bacterial or viral. This has been added to the discussion section as follows:

"Similarly, IFNAR1 has been linked to antiviral immunity more generally as well, hence specificity to flu infections cannot be concluded with complete certainty."

4) As previously described, this study describes a strong association with DQA1*01:02~DQB1*06:02 (DQ0602) haplotype. This haplotype is in linkage disequilibrium with HLA-DRB1*15:01 or HLA-DRB5*01:01. Have you checked the association with these alleles in your analysis?

R3.4. Although this is not always known, having worked for over 20 years on this topic, we have extensive knowledge of HLA haplotypes and their role in narcolepsy. In this context, imputing DRB5 was just not necessary. DRB5*01:01 correlates perfectly with DRB1*15:01 or DRB1*15:03 across all population tested (although null allele can rarely be observed) (see for example the recent papers by Creary et al. (Creary et al., 2019; Creary et al., 2021) (see particularly table S14). Further, DRB5 is centromeric to DRB1*15:01/03 and DRB1*15:01/03 is less associated with narcolepsy than DQA1*01:02~DQB1*06:02 which is telomeric to DRB1*15:01/03.

In 1992, we believe (according to Neil Risch) we were the first to ever use transethnic mapping to show that DQB1*06:02 rather than DRB1*15:01 (or DRB1*15:03 in Africans) was the gene associated with narcolepsy across all ethnic groups (Matsuki et al., 1992). Indeed, in African Americans, the relatively common DRB3*02:02~DRB1*11:01~DQA1*01:02~DQB1*06:02 is as strongly associated with narcolepsy. Similarly, less common haplotypes carrying DQA1*01:02~DQB1*06:02 (DQ0602) in cis of DRB1*03:01, DRB1*08:06 or exceptionally rare recombinants with DRB1*12:02, DR8del and other

alleles have been found to be enriched in narcolepsy, which is 98% associated with DQA1*01:02~DQB1*06:02 (and not DRB1*15 or DRB5*01:01) (Mignot et al., 1997; Mignot et al., 2001; Peraita-Adrados et al., 1999; Peter et al., 1978). These other DRB1 alleles have completely different associated “other” DRB gene/alleles.

One could however have wonder if DRB1*15:01 (and by association DRB5*01:01) *in addition* to DQA1*01:02~DQB1*06:02 could not have played a role (since in Caucasians and most Asians, it is 100% associated). That coffin was nailed when we studied South Chinese Asians, where a frequent haplotype is DRB5*01:01~DRB1*15:01~DQA1*01:02~DQB1*06:01 (an ancestral recombinant between DRB1 and DQA1) exists. In this population, we found no cases of narcolepsy in the presence of DRB1*15:01 without DQA1*01:02~DQB1*06:02 (Han et al., 2012). Again, this indicates that DQA1*01:02~DQB1*06:02 is primary and that there are no detectable effects in DRB1. We even studied some of these very rare (9) cases without DQA1*01:02~DQB1*06:02 and with documented low CSF hypocretin, and did not find anything striking, although a rare DP allele, DPB1*0901, was over-represented in 5 of 9 subjects (Han et al., 2014). Finally, as shown in the text, it is remarkable that other DQA1*01 and DQB1*05/06 allele can compete with DQA1*01:02~DQB1*06:02 in trans to reduce the genetic effect of DQ0602 by cross-heterodimerization (Han et al., 2012; Ollila et al., 2015). This again indicates that the entire DQA1*01:02~DQB1*06:02 (DQ0602) heterodimer and its peptide binding property is critical.

*“The universal genetic association is especially clear for HLA-DQ0602, as it is found with different nearby located HLA-DRB1 alleles: DRB1*15:01 in individuals of primary European (Europe and USA) and Asian (China, Korea, Japan and India) descent, but DRB1*15:03 or DRB1*11:01 in individuals of primary Africa descent. The primacy of DQ0602 over DRB1*15:01 (and thereby DRB5, as LD is complete) is also demonstrated by the fact that the DRB1*15:01~DQA1*01:02~DQB1*06:01 haplotype is not associated with narcolepsy in South China and by the fact additional DQ effects are mostly mediated by DQA1 alleles that interact in trans with DQB1*06:02 (i.e. DQA1*01:01 and DQB1*01:03).”*

Minor comments:

1) In the paragraph " Variants involved in response to influenza" some references to the literature are missing

R3.minor1: We have added the references.

2) Line 301: Fig 1C (not B)

R3.minor2: We have corrected this.

3) Many supplementary tables are cut and consequently the content is not easily readable.

R3.minor3: The supplementary tables have been updated along with other supplementary materials. Thank you for pointing this out.

Bibliography

Barateau, L., Lopez, R., Arnulf, I., Lecendreux, M., Franco, P., Drouot, X., Leu-Semenescu, S., Jaussent, I., and Dauvilliers, Y. (2017). Comorbidity between central disorders of hypersomnolence and immune-based disorders. *Neurology* 88, 93-100.

Chen, T.Y., Su, V.Y., Lee, C.H., Chung, C.H., Tsai, C.K., Peng, C.K., Lai, H.C., Chien, W.C., and Tzeng, N.S. (2021). The Association Between Asthma and Narcolepsy: A Nationwide Case-Control Study in Taiwan. *Nat Sci Sleep* 13, 1631-1640.

Creary, L.E., Gangavarapu, S., Mallempati, K.C., Montero-Martín, G., Caillier, S.J., Santaniello, A., Hollenbach, J.A., Oksenberg, J.R., and Fernández-Viña, M.A. (2019). Next-generation sequencing reveals new information about HLA allele and haplotype diversity in a large European American population. *Hum Immunol* 80, 807-822.

Creary, L.E., Sacchi, N., Mazzocco, M., Morris, G.P., Montero-Martin, G., Chong, W., Brown, C.J., Dinou, A., Stavropoulos-Giokas, C., Gorodezky, C., *et al.* (2021). High-resolution HLA allele and haplotype frequencies in several unrelated populations determined by next generation sequencing: 17th International HLA and Immunogenetics Workshop joint report. *Hum Immunol* 82, 505-522.

Feketeova, E., Tormasiova, M., Klobučníková, K., Durdik, P., Jarcuskova, D., Benca, M., and Vitkova, M. (2020). Narcolepsy in Slovakia - Epidemiology, clinical and polysomnographic features, comorbid diagnoses: a case-control study. *Sleep Med* 67, 15-22.

Han, F., Lin, L., Li, J., Dong, S.X., An, P., Zhao, L., Liu, N.Y., Li, Q.Y., Yan, H., Gao, Z.C., *et al.* (2012). HLA-DQ association and allele competition in Chinese narcolepsy. *Tissue Antigens* 80, 328-335.

Han, F., Lin, L., Schormair, B., Pizza, F., Plazzi, G., Ollila, H.M., Nevsimalova, S., Jennum, P., Knudsen, S., Winkelmann, J., *et al.* (2014). HLA DQB1*06:02 negative narcolepsy with hypocretin/orexin deficiency. *Sleep* 37, 1601-1608.

Jacob, L., Leib, R., Ollila, H.M., Bonvalet, M., Adams, C.M., and Mignot, E. (2015). Comparison of Pandemrix and Arepanrix, two pH1N1 AS03-adjuvanted vaccines differentially associated with narcolepsy development. *Brain Behav Immun* 47, 44-57.

King, L.E., Jr., Eastham, A.W., Curcio, N.M., and Schmidt, A.N. (2010). A potential association between alopecia areata and narcolepsy. *Arch Dermatol* 146, 677-679.

Kornum, B.R., Knudsen, S., Ollila, H.M., Pizza, F., Jennum, P.J., Dauvilliers, Y., and Overeem, S. (2017). Narcolepsy. *Nat Rev Dis Primers* 3, 16100.

Martinez-Orozco, F.J., Vicario, J.L., De Andres, C., Fernandez-Arquero, M., and Peraita-Adrados, R. (2016). Comorbidity of Narcolepsy Type 1 With Autoimmune Diseases and Other Immunopathological Disorders: A Case-Control Study. *J Clin Med Res* 8, 495-505.

Matsuki, K., Grumet, F.C., Lin, X., Gelb, M., Guilleminault, C., Dement, W.C., and Mignot, E. (1992). DQ (rather than DR) gene marks susceptibility to narcolepsy. *Lancet* 339, 1052.

Mignot, E., Kimura, A., Lattermann, A., Lin, X., Yasunaga, S., Mueller-Eckhardt, G., Rattazzi, C., Lin, L., Guilleminault, C., Grumet, F.C., *et al.* (1997). Extensive HLA class II studies in 58 non-DRB1*15 (DR2) narcoleptic patients with cataplexy. *Tissue Antigens* 49, 329-341.

Mignot, E., Lin, L., Rogers, W., Honda, Y., Qiu, X., Lin, X., Okun, M., Hohjoh, H., Miki, T., Hsu, S., *et al.* (2001). Complex HLA-DR and -DQ interactions confer risk of narcolepsy-cataplexy in three ethnic groups. *Am J Hum Genet* 68, 686-699.

Mullarkey, M.E., Stevens, A.M., McDonnell, W.M., Loubière, L.S., Brackensick, J.A., Pang, J.M., Porter, A.J., Galloway, D.A., and Nelson, J.L. (2005). Human leukocyte antigen class II alleles in Caucasian women with primary biliary cirrhosis. *Tissue Antigens* 65, 199-205.

Nigam, G., Pathak, C., and Riaz, M. (2016). Alopecia areata and narcolepsy: a tale of obscure autoimmunity. *BMJ Case Rep* 2016.

Ollila, H.M., Ravel, J.M., Han, F., Faraco, J., Lin, L., Zheng, X., Plazzi, G., Dauvilliers, Y., Pizza, F., Hong, S.C., *et al.* (2015). HLA-DPB1 and HLA class I confer risk of and protection from narcolepsy. *Am J Hum Genet* 96, 136-146.

Peraita-Adrados, R., Ezpeleta, D., Balas, A., and Vicario, J.L. (1999). Narcolepsy-cataplexy syndrome associated with DRB1*0806-DQB*0602 haplotype in a Caucasian patient. *Sleep Res Online* 2, 29-31.

Peter, H.H., Deutschmann, K.E., Deinhardt, J., and Deicher, H. (1978). Value of adjuvant therapy with bacille Calmette Guerin (BCG) or dimethyl triazeno imidazole carboximide (DTIC) in the control of minimal residual disease in stage II melanoma. *Recent Results Cancer Res* 68, 367-374.

Simmons, K.M., Mitchell, A.M., Alkanani, A.A., McDaniel, K.A., Baschal, E.E., Armstrong, T., Pyle, L., Yu, L., and Michels, A.W. (2020). Failed Genetic Protection: Type 1 Diabetes in the Presence of HLA-DQB1*06:02. *Diabetes* 69, 1763-1769.

Ye, C.J., Chen, J., Villani, A.C., Gate, R.E., Subramaniam, M., Bhangale, T., Lee, M.N., Raj, T., Raychowdhury, R., Li, W., *et al.* (2018). Genetic analysis of isoform usage in the human anti-viral response reveals influenza-specific regulation of ERAP2 transcripts under balancing selection. *Genome Res* 28, 1812-1825.

REVIEWER COMMENTS

Reviewer #1 (Remarks to the Author):

The data are certainly novel and important, and their presentation has clearly improved. Still, a large number of inconsistencies and/or approximations remains hampering the full understanding of what exactly has been done.

I guess the authors ought to read carefully and critically their manuscript including the supplementary material to make the final changes needed for publication in a Nature journal.

Among others, here are some areas that need clarification:

1. The number of individuals studied sometimes do not add up
 - Page 9 and 12: total 6,073 NT1 cases but 5,848 cases from 10 cohorts and 245 vaccination induced NT1 (add up to 6,093!)
 - Suppl information: clinical cohort of 387 unrelated type 1 narcolepsy (NT1) patients and 1,476 unrelated Japanese controls (N total = 1,971). How do the authors come up with this 1,971 figure?
 - Suppl information: East Asian sample initially included 556 individuals with narcolepsy from China (N=320), Japan (N=1), Taiwan (N=11), and Korea (N=182), as well as individuals of Asian descent living outside Asia (N=48). How do the authors come up with the total figure of 556?
 - Suppl information: This cohort is comprised of previously reported individuals with narcolepsy (N=625 with Affy6, and 182 individuals with Affymetrix 500k) and controls (N=915 with Affy6 and 155 with Affymetrix 550k, total N=1,881). How do the authors come up with this 1,881 figure?
 - Page 18: 130 individuals sequences specifically for both memory CD4+ and CD8+ T cells whereas in Suppl page 15 it is indicated "targeted RNA expression sequencing data from 60 patients and 60 DQ0602 controls"
2. It is still unclear what is represented in Suppl Fig 4 as the legends is confusing "Here we show IFNAR1 locus in narcolepsy and influenza-A infection after H1N1 infection in dendritic cells in Supplementary figure 4". Is it IFNAR1 expression in DC as stated on page 15?
 - page 15-16 how do the authors link IFNAR1 expression in DC and influenza virus uptake and antigen presentation?
3. Page 16, the HLA-A03:01 allele is stated to be predisposing whereas on Table S6 it appears protective.
 - Similarly, the Perforin A91V variant is described as protective on page 22, while this allele appears associated with NT1 on Table S3.
4. The genotypes of TRA locus rs1154155 are T or G on Suppl Fig 9 and 10 whereas on page 34 rs1154155C is indicated. Is there more than two possible variants at this single position?
 - The TRB SNP of interest is named rs1008599 in Suppl Fig 7 and 8 and rs1008955 on page 19. Which one is it?
5. Some Supplementary figures still lack proper legends.
6. Can some Tables be simplified to focus on the relevant information. For instance, on Table 9 what is the use of columns 1, 3, 4, 5.
7. The p value indicated on page 19 first line cannot be > 1.
8. Some references need to be completed/updated such as ref 5, 10.

9. Figure 3: Are T cells a meaningful cellular source of IL-1, IL-18 and IL-12? Which data support this scheme.

Reviewer #2 (Remarks to the Author):

This manuscript reports remarkably large GWAS of >6000 type 1 narcolepsy cases and over 84,000 controls. The results confirm 6 known risk alleles associated with type 1 narcolepsy and discover 7 novel non-MHC associations. Many of these associations transcend ethnicity, and a number are shared with other autoimmune diseases. The findings significantly extend what was known about the pathogenesis of narcolepsy. Overall, the authors have been exceptionally responsive to all reviewer concerns, and have extensively re-written the manuscript such as to make it far more readable, interpretable, and clear than was the original version. I believe this manuscript will be of broad interest and widely cited.

Reviewer #3 (Remarks to the Author):

This reviewer thanks the authors for having addressed my concerns in the revised manuscript and endorses the publication of this paper, which overall provides further evidence to the notion of narcolepsy as an autoimmune disorder.

Minor comments:

The text contains typos/minor English issues in many sentences. Please, fix them throughout the text.

Figure 3:

1) In the figure, IL-1 and IL-18 are shown to be produced by Th1 cells. However, they are known to be mostly released by innate cells (monocytes, DCs and NK) and to consequently influence T cell polarization. Please fix this in the figure.

2) the legend of this figure states that "Activated and primed specific CD4+ cells migrate to the CNS, where they interact with microglia and resident DCs via DQ0602 bound to an influenza-mimic autoimmune-epitope (derived from hypocretin cells), initiating a secondary memory response..." What is the evidence for that? If there is, it should be provided or otherwise consider to replace it with known concepts of narcolepsy immunopathology based on CD4+ and CD8+ T cells infiltration from the periphery and CD8+ T cell mediate neuronal killing by CD8+ T cells (R Bernard-Valnet, PNAS, 2016; Latorre et al. Semin Immunopathol, 2022).

Response to Reviewers

Reviewer #1

Remarks

The data are certainly novel and important, and their presentation has clearly improved. Still, a large number of inconsistencies and/or approximations remains hampering the full understand of what exactly has been done. I guess the authors ought to read carefully and critically their manuscript including the supplementary material to make the final changes needed for publication in a Nature journal.

Reply: We thank the reviewer for looking over our manuscript with a fine-tooth comb and identifying these errors.

Among others, here are some areas that need clarification:

Comment 1

The number of individuals studied sometimes do not add up

1.1

Page 9 and 12: total 6,073 NT1 cases but 5,848 cases from 10 cohorts and 245 vaccination induced NT1 (add up to 6,093!)

Reply: We thank the reviewer for catching up this error. This was merely a typo in the main text. Accordingly, we have corrected the number to 225 throughout the main text.

1.2

Suppl information: clinical cohort of 387 unrelated type 1 narcolepsy (NT1) patients and 1,476 unrelated Japanese controls (N total = 1,971). How do the authors come up with this 1,971 figure?

Reply: The clinical cohort initially included 409 NT1 cases and 1,562 controls, however, a total of 108 individuals (n=22 T1N cases and 86 controls) were removed because of Quality Control (QC) prior to imputation. While the initial cohort adds up to 1,971 individuals, only 1,863 subjects passed filters and QC.

	Cases	Controls	Total
Pre-QC	409	1,562	1,971
Post-QC	387	1,476	1,863

We apologies for the unclarity and have now adjusted the supplementary materials on page 2 to clearly state the initial cohort size of 409 + 1,562 = 1,971 prior to explaining QC, filtering, and subsequent removal of subjects.

Modified text: “The Japanese cohort is a clinical cohort of 409 unrelated type 1 narcolepsy (NT1) patients and 1,562 unrelated Japanese controls (N total = 1,971) genotyped and analyzed in Japan. [...] A total of 108 individuals (n=22 T1N cases and 86 controls) were removed because of Quality Control (QC) prior to imputation. 43,400 SNPs were excluded based on genotyping rate (1% or more missing genotypes). 503,239 variants and 1,863 subjects passed filters and QC.”

1.3

Suppl information: East Asian sample initially included 556 individuals with narcolepsy from China (N=320), Japan (N=1), Taiwan (N=11), and Korea (N=182), as well as individuals of Asian decent living outside Asia (N=48). How do the authors come up wwith the total figure of 556?

Reply: We apologies for the unclarity. 562 is the original number of subjects, however, 34 of these were removed due to QC and thus 528 were carried forward for the final analysis (see supplementary table 6). We have edited the description of this cohort accordingly.

1.4

Suppl information: This cohort is comprised of previously reported individuals with narcolepsy (N=625 with Affy6, and 182 individuals with Affymetrix 500k) and controls (N=915 with Affy6 and 155 with Affymetrix 550k, total N=1,881). How do the authors come up with this 1,881 figure?

Reply: Once again we apologize for the unclarity of our description. The total number of individuals included post QC were 1,889. This is slightly higher than the total number used in Hallmayer et al., 2009; the reason for this is improved QC, which has allowed us to carry forward more samples for analysis than we did in this earlier paper from our group. In specific, we were able to rescue more individuals from Affymetrix 500 (167 rather than just 155). We have updated the description accordingly.

1.5

Page 18: 130 individuals sequences specifically for both memory CD4+ and CD8+ T cells whereas in Suppl page 15 it is indicated "targeted RNA expression sequencing data from 60 patients and 60 DQ0602 controls"

Reply: we thank the reviewer for thorough investigation; this was a typo and we have now corrected the number to 120 on page 18.

Comment 2

It is still unclear what is represented in Suppl Fig 4 as the legends is confusing

Reply: We agree. The legend was changed as follows:

LD was examined using LocusZoom (Pruim et al., 2010). The LocusZoom plot shows variants of the IFNAR1 locus in narcolepsy (left) and variants of the IFNAR1 locus associated with IFNAR1 expression, following influenza-A H1N1 infection in dendritic cells (right). We thereby show that the leading narcolepsy risk variant rs2409487 is in perfect LD with the lead in Influenza-A infection eQTL (rs6517159) in dendritic cells and colocalization analysis suggests shared signal. We have updated the legend of the figure accordingly. Further, we corrected the corresponding section in the main text to the following:

“We therefore examined IFNAR1 expression in DC following H1N1 infection (PR8 delta NS1) and interferon stimulation, finding that the same SNP (rs2096464) or a highly linked SNP (rs6517159, $r^2 = 0.93$, Supplementary Fig. 4) are major eQTLs for these effects ($p = 1.92 \times 10^{-25}$, $\beta = 0.140$ rs2096464 for flu infection; $p = 10^{-33}$, $\beta = 0.215$ rs6517159 following interferon stimulation).”

Comment 3

Page 16,

3.1

The HLA-A03:01 allele is stated to be predisposing whereas on Table S6 it appears protective.

Reply: Thank you for pointing this out. We found an error in Table 6. Specifically, some of the class I association had been controlled for by A*11:01, which changed the direction of association for other strongly linked class I alleles. Since the HLA class I associations are rather weak, we decided to not control for top HLA class I allele A11:01. Based on LD across ethnic groups, HLA-A11:01 is the most likely true signal, thus it made sense, but effects of B51:01, B35:01 and B35:03 are also possible and not to adjust is more conservative (see Ollila et al., 2015). We “updated supplementary table 6 and clarified the main text as follows:

“Less significant associations were found with risk increasing alleles at class II DPB105:01 and at HLA class I with A11:01, B51:01, B35:01 and B35:03, and risk reducing allele A03:01 ($p < 0.01$, Table S6).”

3.2

Similarly, the Perforin A91V variant is described as protective on page 22, while this allele appears associated with NT1 on Table S3.

Reply: Our sincere apologies to the reviewer, but we could not identify which discrepancy he is trying to outline. rs78325861 is noted to be protective in both the main text on p.22, in supplementary table 3 and most notably also in main table 1. The beta is consistently reported to be negative i.e. protective with the rare allele rs78325861G, an allele linked with rs35947132A. Please tell us if we are mistaken or misunderstood your comment.

Comment 4

The genotypes of TRA locus rs1154155 are T or G on Suppl Fig 9 and 10 whereas on page 34 rs1154155C is indicated. Is there more than two possible variants at this single position? The TRB SNP of interest is named rs1008599 in Suppl Fig 7 and 8 and rs1008955 on page 19. Which one is it?

Reply: Regarding the genotypes of TRA locus rs1154155:

We are sorry for this error which occurred due to the nomenclature of rs1154155 and rs1483979, which both involve a G allele with opposite effects. Specifically, the rarer Caucasian G allele of rs1154155 predisposes to NT1, unlike the more frequent T allele, and rs1154155G is linked to the rarer C allele of rs1483979, which is associated with an L to F substitution (alternate allele rs1483979G (L)). Of further note, the effect is the same in individuals of Asian descent, however, the frequency of these alleles is reversed in Asians, which makes it easy to make a mistake. Accordingly, we have now corrected the legend on page 34 to rs1154155G, and checked accuracy all through.

Regarding rs1008599:

Our apologies, this is a typo in the main text, which we have now corrected. The main text thus now states rs1008599 as well, consistent with supplementary figures 7 & 8 and location in TCRB.

Comment 5

Some Supplementary figures still lack proper legends.

Reply: We have added further details and clarifications to the legends in several places and hope this is to the satisfaction of the reviewer.

Comment 6

Can some Tables be simplified to focus on the relevant information. For instance, on Table 9 what is the use of columns 1, 3, 4, 5.

Reply: We were unclear on what exactly ought to be deleted in this supplementary table and as this is supplementary materials, we erred towards listing more information.

Comment 7

The p value indicated on page 19 first line cannot be > 1.

Reply: We are incredibly thankful for such thorough examination of our manuscript! This was a typing mistake and should have listed 2.29×10^{-10} , instead, which we have now corrected in the main text accordingly.

Comment 8

Some references need to be completed/updated such as ref 5, 10.

Reply: We updated the references.

Comment 9

Figure 3: Are T cells a meaningful cellular source of IL-1, IL-18 and IL-12? Which data support this scheme.

Reply: We have completely re-done this figure to better reflect our pathophysiological model. In response to criticisms raised by both this reviewer and reviewer 3, we also incorporated more recent literature in this figure. This now includes citations of recent works by Luo et al., 2022, Bernard-Valent et al., 2022 and Pedersen et al., 2019.

Reviewer #2

Remarks

This manuscript reports remarkably large GWAS of >6000 type 1 narcolepsy cases and over 84,000 controls. The results confirm 6 known risk alleles associated with type 1 narcolepsy and discover 7 novel non-MHC associations. Many of these associations transcend ethnicity, and a number are shared with other autoimmune diseases. The findings significantly extend what was known about the pathogenesis of narcolepsy. Overall, the authors have been exceptionally responsive to all reviewer concerns, and have extensively re-written the manuscript such as to make it far more readable, interpretable, and clear than was the original version. I believe this manuscript will be of broad interest and widely cited.

Reply: Thank you so much for you comment and review in improving this manuscript.

Reviewer #3

Remarks

This reviewer thanks the authors for having addressed my concerns in the revised manuscript and endorses the publication of this paper, which overall provides further evidence to the notion of narcolepsy as an autoimmune disorder.

Comment 1

Minor comments: The text contains typos/minor English issues in many sentences. Please, fix them throughout the text.

Reply: We thank the reviewer for pointing this out. We have spotted some mistakes that we missed during previous editing and fixed these. We have also asked a native English speaker for a final review and editing.

Comment 2

Figure 3:

2.1

In the figure, IL-1 and IL-18 are shown to be produced by Th1 cells. However, they are known to be mostly released by innate cells (monocytes, DCs and NK) and to consequently influence T cell polarization. Please fix this in the figure.

Reply: Please see our response to reviewer #1 comment 9. The figure was entirely redone.

2.2

the legend of this figure states that “Activated and primed specific CD4+ cells migrate to the CNS, where they interact with microglia and resident DCs via DQ0602 bound to an influenza-mimic autoimmune-epitope (derived from hypocretin cells), initiating a secondary memory response...” What is the evidence for that? If there is, it should be provided or otherwise consider to replace it with known concepts of narcolepsy immunopathology based on CD4+ and CD8+ T cells infiltration from the periphery and CD8+ T cell mediate neuronal killing by CD8+ T cells (R Bernard-Valnet, PNAS,2016; Latorre et al. Semin Immunopathol,2022).

Reply: Please see our response to reviewer #1 comment 9. The figure was entirely redone.

REVIEWERS' COMMENTS

Reviewer #1 (Remarks to the Author):

The authors did a fine job and satisfactorily addressed my previous concerns.